# Compensatory growth and recovery of cartilage cytoarchitecture after transient cell death in fetal mouse limbs

Chee Ho H'ng[1], Shanika L. Amarasinghe[1,2,5], Boya Zhang [1,5], Hojin Chang [1,4,5], Xinli Qu[1], David R. Powell [2] & Alberto Rosello-Diez [1,3] ✉

A major question in developmental and regenerative biology is how organ size and architecture are controlled by progenitor cells. While limb bones exhibit catch-up growth (recovery of a normal growth trajectory after transient developmental perturbation), it is unclear how this emerges from the behaviour of chondroprogenitors, the cells sustaining the cartilage anlagen that are progressively replaced by bone. Here we show that transient sparse cell death in the mouse fetal cartilage is repaired postnatally, via a two-step process. During injury, progression of chondroprogenitors towards more differentiated states is delayed, leading to altered cartilage cytoarchitecture and impaired bone growth. Then, once cell death is over, chondroprogenitor differentiation is accelerated and cartilage structure recovered, including partial rescue of bone growth. At the molecular level, ectopic activation of mTORC1 correlates with, and is necessary for, part of the recovery, revealing a specific candidate to be explored during normal growth and in future therapies.

Developmental robustness, i.e. the ability to achieve a relatively normal body plan despite genetic and environmental perturbations during development[1], plays a key role in fitness and natural selection, but the underlying mechanisms are poorly understood. When it involves recovery of a normal growth trajectory after transient growth impairment, the phenomenon is referred to as catch-up growth[2], which is related to—but distinct from—regeneration. In adult tissue regeneration, final organ mass had already been established before the injury, and there needs to be recovery of the lost mass. On the other hand, catch-up growth requires the recovery of potential mass, that is, the mass that would have been produced had the perturbation not happened. At the conceptual level, one potential scenario is that stem/progenitor cells can monitor a parameter directly or indirectly related to current organ size, and that this parameter dictates cell behaviours, but the potential mechanisms for this scenario remain unexplored. It is also unclear why compensatory/repair mechanisms are in general

more powerful during fetal and perinatal stages than at juvenile stages. Thus, shedding light on this age-related decline will have a transformative effect on future regenerative therapies. Here, we addressed these fascinating questions using the vertebrate limb as a model, focusing on the long bones.

Growth of the long bones takes place by endochondral ossification, whereby a transient cartilage scaffold is progressively replaced by bone[3,4]. This replacement starts at the centre of the fetal skeletal elements, forming the primary ossification centre (Fig. 1a). In mice, secondary ossification centres form ~1 week postnatally at both ends of the skeletal elements so that, at each end, a cartilage disc (the growth plate) gets sandwiched between the two ossification centres. Cells in the growth plate (chondrocytes) traverse subsequent differentiation states from distal positions towards the bone centre: the resting zone (RZ) contains a pool of round quiescent progenitors, which in the proliferative zone (PZ) transition to flat proliferative chondrocytes

[1]Australian Regenerative Medicine Institute, Monash University, Clayton 3800 VIC, Australia. [2]Bioinformatics Node – Monash Genomics and Bioinformatics Platform, Monash University, Clayton 3800 VIC, Australia. [3]Department of Physiology, Development and Neuroscience, University of Cambridge, Cambridge, United Kingdom. [4]Present address: Biological Optical Microscopy Platform, Faculty of Medicine, Dentistry & Health Sciences. The University of Melbourne, Parkville 3010 VIC, Australia. [5]These authors contributed equally: Shanika L. Amarasinghe, Boya Zhang, Hojin Chang.
✉e-mail: alberto.rosellodiez@monash.edu; ar2204@cam.ac.uk

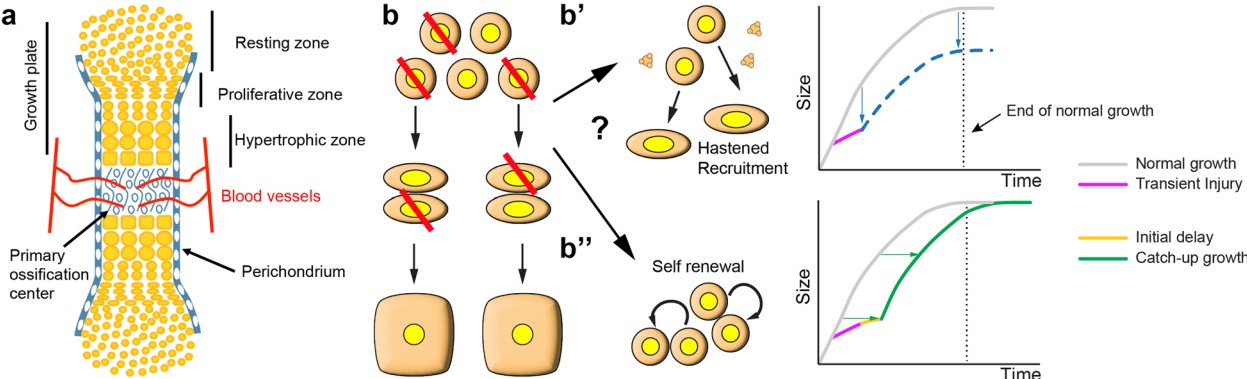

**Fig. 1 | Model and approach to study fetal cartilage repair. a** Schematic of a fetal skeletal element undergoing endochondral ossification. **b–b"** Potential outcomes of mosaic cell ablation (red diagonal lines) in the fetal cartilage. Chondrocyte progenitors can either be immediately recruited into the proliferative population (**b'**), using up their potential faster (eventually failing to catch-up). Alternatively, chondroprogenitors first self-renew (**b"**), as shown by Oichi et al.[28], and then continue proliferating, leading to catch-up after a delay.

stacked in longitudinal columns, and these cells then enlarge to form hypertrophic chondrocytes (HTCs, Fig. 1a). HTCs coordinate bone formation, vascularization and cartilage resorption by attracting and communicating with the blood vessels that invade and degrade the cartilage from the surrounding tissues, accompanied by osteoblast precursors[5–7]. Many HTCs eventually die after laying down the extracellular matrix that will be replaced by bone matrix, which is produced by osteoblasts. The latter can also arise from the transdifferentiation of HTCs that do not die[8–11], but in this case HTCs do not seem to contribute as much to bone elongation as they do to bone mass accrual[12–15].

The growth plate is regulated by multiple local and systemic signals[3,4], and one of the properties that are tightly controlled is its cytoarchitecture, including the relative heights of the resting, proliferative and hypertrophic zones. Indeed, the proportion between the different growth-plate zones is a species-specific trait that correlates with bone size and growth rate, and that changes with age and anatomical location[16,17]. Yet, how this proportion is established is only partially understood. The size of the proliferative zone depends on a negative feedback loop between parathyroid hormone related peptide (PTHrP), produced by resting chondrocytes, and Indian hedgehog (IHH), produced by prehypertrophic ones (Fig. 1a). This feedback loop acts as a rheostat that couples proliferation and differentiation to maintain a certain growth-plate size despite changes in cartilage production and degradation rates[18]. Regarding the cellular parameters affecting the actual growth rate, most of bone elongation has been shown to be contributed by the hypertrophic zone, via both cell enlargement and production of extracellular matrix[19]. Consequently, in growth plates whose HTCs reach very big sizes, changes in chondrocyte proliferation also exert a major effect on bone elongation, as they change the number of HTCs produced in a given time period[20].

One approach to gain insight into the regulation of bone growth is to create structural defects in the cartilage and study the recovery (or lack thereof) of an adequate cartilage cytoarchitecture. Given that the mechanisms of fetal repair are more powerful than postnatal ones[21], we posited that using this approach in fetal bones would be more informative than in postnatal ones. To gain insight into how bone growth rate is controlled, some labs (including ours) have studied the process of catch-up growth[1,2]. Three decades ago, Baron et al. showed that catch-up growth can happen after a local perturbation of bone growth[22], suggesting that cartilage-intrinsic, and not systemic mechanisms, underlie the recovery. Subsequent studies refined this model, which became referred to as 'autonomous'[23–26]. According to this model, chondrocyte progenitors have an intrinsically limited proliferative potential that is partially used up with every round of cell

division until they become senescent. At this point, the growth plate is consumed faster than it is replaced, and bone growth ceases. In this framework, catch-up growth is explained as follows: transient impairment of chondrocyte progenitor proliferation preserves their potential, so that when the insult is lifted, their 'age' is lower than their chronological age. Therefore, cartilage growth resumes at a speed associated with younger stages and lasts for a longer-than-normal period, until chondrocyte progenitors become senescent. This means that the growth trajectory is shifted horizontally towards a more advanced age, and the actual catch-up happens towards the normal end of the growth period. This model makes two strong predictions: (1) if the insult is mosaic (i.e., in a salt & pepper pattern), spared chondrocytes should keep behaving normally and not participate in the catch-up; (2) if chondrocytes are killed instead of arrested, there should not be catch-up, as the surviving cells keep using up their proliferative potential normally. The second prediction was tested in this study, as we shall show below. The first prediction was tested by us in a previous study[27], by inducing continuous mosaic cell-cycle arrest in chondrocytes, as opposed to the transient but widespread arrest induced in the study by Baron et al.[22]. Unexpectedly, the non-arrested chondrocytes showed enhanced proliferation rate, compensating for the proliferative arrest of their neighbours. The most parsimonious interpretation of this result is that the spared chondrocytes showed a cell-nonautonomous response, likely mediated by cell-cell communication and hence not compatible with a strict interpretation of the autonomous model. However, since we could not trace the fate of the arrested chondrocytes, the remote possibility existed that some of them escaped the cell-cycle arrest and participated in the compensation, as suggested by the autonomous model.

To avoid the abovementioned caveat and test the autonomous model of catch-up growth in a more definitive way, in this study we killed chondrocytes instead of arresting them, such that they could not possibly participate in the potential recovery. We induced transient mosaic cell death in the cartilage of the left fetal long bones (Fig. 1b), with the right limb as internal control. We predicted two potential scenarios at the level of resting chondrocytes. In one scenario, the remaining chondroprogenitors accelerate their recruitment towards the proliferative pool, but some are used to fill-in the gaps left by the dying ones, instead of being used to promote elongation, leading to a progressively increasing length difference with the control limb (Fig. 1b'). In the alternative scenario, the injury could induce self-renewal of some of the spared chondroprogenitors at the expense of the PZ pool, leading to an initial growth delay, followed by elongation at the rate of a younger animal, and catch-up after a longer time (Fig. 1b"). As mentioned above, the latter scenario was recently found in the case of catch-up growth induced by transient diet restriction[28]. In

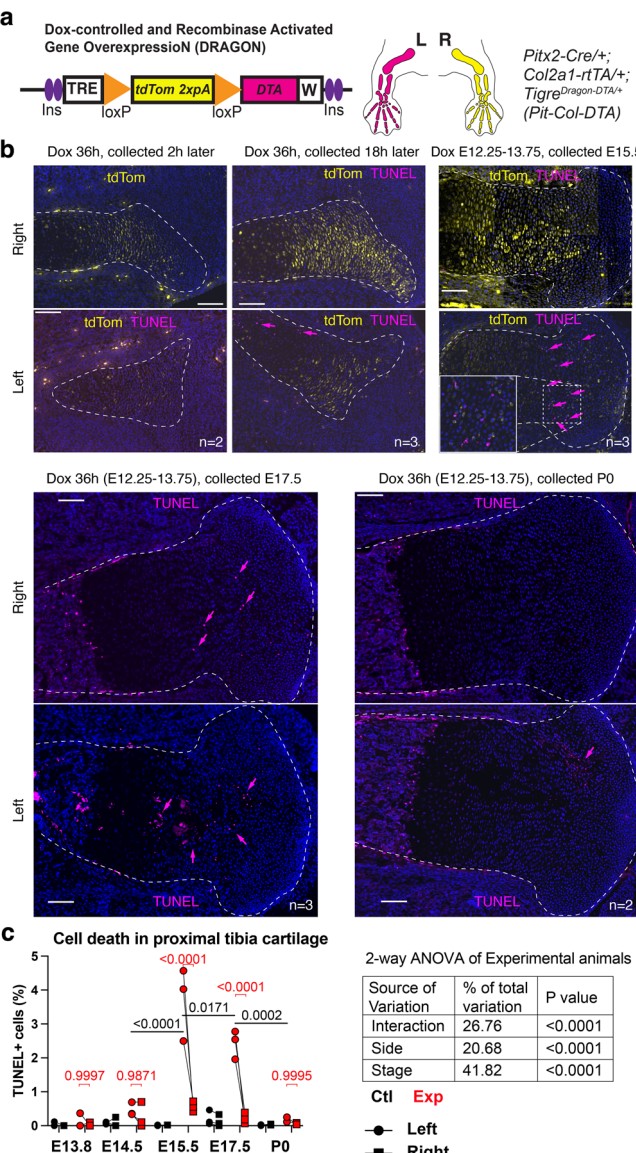

**Fig. 2 | A model of transient unilateral cell death in the fetal cartilage. a** Genetic combination to achieve unilateral, cartilage-specific, inducible and reversible DTA expression. **b** Examples of the tdTomato expression and cell death (TUNEL, arrows) achieved in the cartilage (dashed lines) after the indicated Dox treatments. Scale bars represent 100 μm. **c** Quantification of TUNEL⁺ cells at the indicated stages/locations. Two-way ANOVA table is shown, with the post-hoc multiple comparison tests (across sides and across stages in Exp cartilage) shown above the graphs.

contrast, we unexpectedly found that a hybrid compensatory growth took place via two mechanisms: (1) replenishment of resting chondrocytes due to a transient bias towards self-renewal, followed by accelerated transition to the proliferative pool, but insufficient to gain back the lost size; (2) transiently increased HTC size. These responses were associated with increased phosphorylated ribosomal protein S6 (p-S6) signal, partially dependent on mechanistic target of rapamycin complex 1 (mTORC1) signalling. Moreover, the compensatory response led to an almost perfect recovery of the zone proportions in the injured growth plates, a process which was impaired by inhibition of mTORC1 signalling, in line with the modulatory effect that p-S6 has been recently shown to play in skin wound healing[29]. These results suggest a model of catch-up growth that integrates the available data and reveal molecular candidates that could be targeted to improve or extend the reparative phase.

## Results

### A model of acute unilateral cell death in the fetal limb cartilage

To induce transient cell death in the cartilage of the left developing limbs, we utilized a double conditional mouse strain (*Tigre^Dragon-DTA*, *Dragon-DTA* hereafter) that we previously generated[30], in which expression of attenuated diphtheria toxin fragment A (DTA) requires exposure to previous or current Cre activity as well as activated tetracycline-responsive transactivator (tTA), or the reverse version rtTA. Cre activity is required to prime DTA expression by removing a floxed tdTom-STOP cassette, and (r)tTA drives transcription of either tdTomato (in the unrecombined allele) or DTA (in the recombined one) from a Tet-responsive element (TRE) located upstream of the transgene (Fig. 2a). Since the activity of (r)tTA can be controlled by the Tetracycline analogue Doxycycline (Dox, which activates rtTA and inhibits tTA), the model is inducible and reversible[30]. We crossed males homozygous for this allele with females homozygous for *Pitx2-ASE-Cre*[31] (*Pitx2-Cre* hereafter) and hemizygous for *Col2a1-rtTA*[32], to generate the *Pit-Col-DTA* model. The *Pitx2-Cre* allele drives Cre expression in the left lateral plate mesoderm (which gives rise to the limb mesenchyme), while *Col2a1-rtTA* drives rtTA expression in the *Col2a1*-expressing chondrocytes[32], so that their activities intersect in the cartilage of the left limbs, killing chondrocytes in those skeletal elements[30]. To achieve a brief peak of cell death, but intense enough to generate an obvious left-right asymmetry, we first tested several Dox regimes in the drinking water, varying the concentration and duration of treatment. 0.5 mg/ml Dox given from embryonic day (E) 12.5–E15.5 caused limb asymmetry in some specimens, but the penetrance was low unless Dox concentration was increased to 1 mg/ml. When 1 mg/ml Dox was given for 12 h and the embryos collected right after, incipient tdTomato signal was detected in the right cartilage, indicating that the TRE had been active for a few hours, although cell death (assessed via TUNEL staining) was not detected yet (Supplementary Fig. 1a). Interestingly, when Dox was given at E12.25 and the embryos collected 30 h later (at E13.5), we detected clear expression of *DTA* in the left cartilage, but still not cell death (Supplementary Fig. 1b). Similarly, when Dox was given from E12.25 to E13.75 (i.e., for 36 h) and the embryos collected 2–4 h later (E13.8), clear tdTomato but little TUNEL signals were detected (Fig. 2b, c). After a similar 36-h Dox pulse, the first clear indication of cell death in the left cartilage was found 18 h later (~E14.5), and especially expanded 42 h after Dox withdrawal (~E15.5, Fig. 2b, c). The levels of cell death were still quite high 2 days later, at E17.5, but almost null by E19.5, i.e. postnatal day 0 (P0, Fig. 2b, c). The results thus show that transient Dox treatment in the *Pit-Col-DTA* model indeed triggers acute cell death. Of note, the right experimental limb (i.e. the internal control) showed just a few DTA⁺ cells at E13.5 and E14.5 (Supplementary Fig. 1b, arrowheads) and a minor trend towards increased cell death as compared to absolute controls at E15.5 (Fig. 2c). This was somewhat expected, as we previously showed that *Pitx2-Cre* exhibits some minor activity in the right limb too[27].

### Cartilage is almost completely repaired 1 week after injury

After confirming the transient peak of cell death in the *Pit-Col-DTA* model, we analysed the effect on cartilage integrity and cytoarchitecture. Haematoxylin and eosin staining of the proximal growth plate at multiple stages showed obvious gaps in the E17.5–E18.5 left experimental cartilage (Exp L), spreading across the resting and proliferative zones (Fig. 3a and Supplementary Fig. 2a). By P0, the damage had expanded partially or completely to the HZ, showing big acellular patches, filled only with extracellular matrix (Fig. 3b). Most remarkably, however, by P3–P5 most of the damage in the resting zone had healed, and the gaps in other zones were less obvious in most cases (Fig. 3c, d. Only one out of six Exp L tibia showed empty patches bigger than four cell diameters). By P7, no cartilage gaps were detected (Supplementary Fig. 2b). However, some long-term effects of the acute injury were still detectable at P7. For example, the whole cartilage was

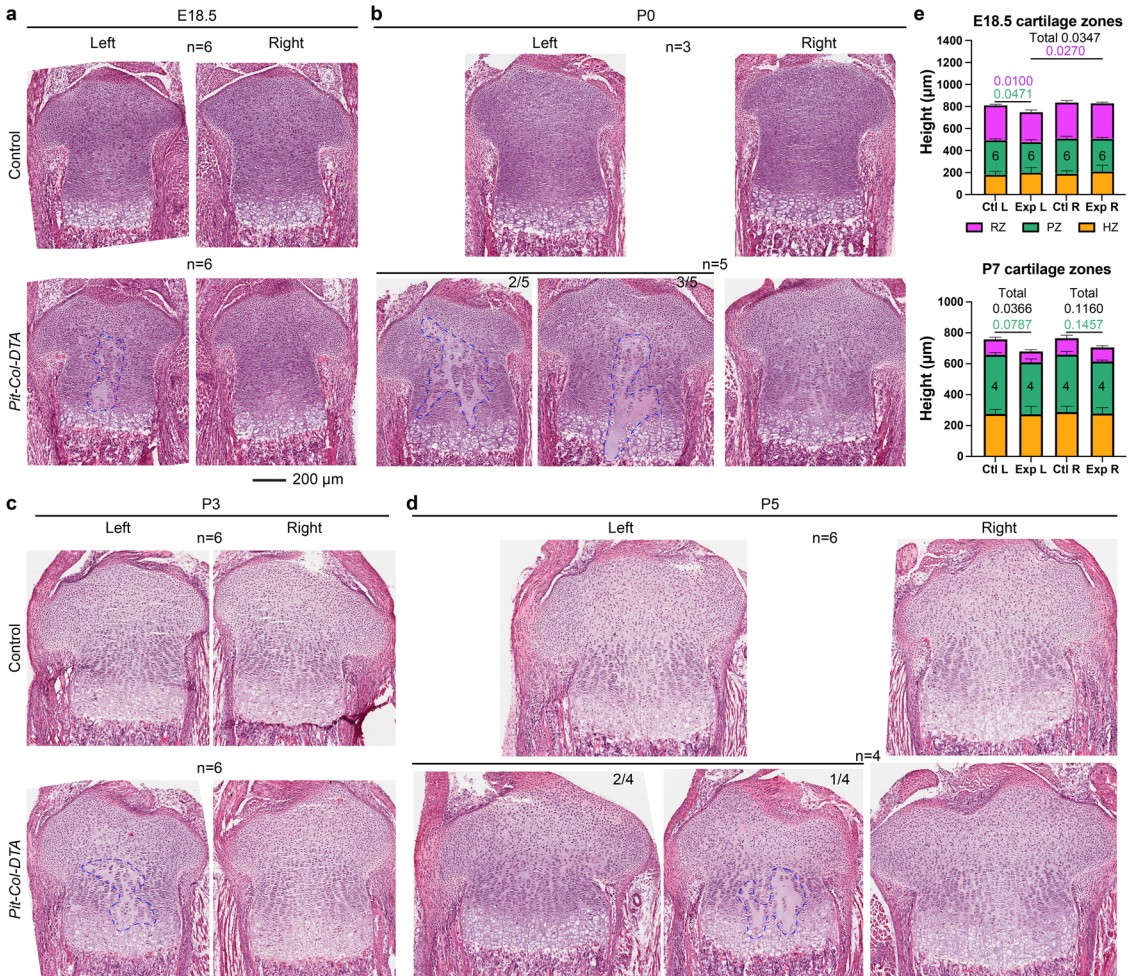

**Fig. 3 | After transient cell death, cartilage cytoarchitecture is repaired within 1 week. a–d** H&E staining on sections of the proximal tibia at the indicated embryonic (E) and postnatal (P) stages. Dashed lines delineate acellular cartilage areas. Scale bar, 200 μm; applies to panels (**a–d**). **e** Quantification of the maximum length of each cartilage zone (RZ/PZ/HZ, resting/proliferative/hypertrophic) at the indicated stages, categorized by genotype and side. Number of samples as indicated inside the green zone of the bar graphs. Error bars are SD. Statistical analyses are two-way ANOVAs and Sidak's post-hoc multiple comparisons tests, performed at each stage for all zones at the same time, in a pairwise manner (CtlL-ExpL, CtlR-ExpR, CtlL-CtlR, ExpL-ExpR).

slightly shorter in the Exp L tibia as compared to controls, although no differences between Exp L and Exp R were found (Fig. 3e). Lastly, we noticed that formation of the secondary ossification centre (SOC) was delayed in the P7 Exp L proximal tibia, suggesting that the cartilage repair process entails an overall delay of the endochondral ossification process (Supplementary Fig. 2c).

**Transient unilateral cell death in the developing limb cartilage leads to limb asymmetry, followed by relative catch-up growth**
We next analysed the effect of transient unilateral cartilage cell-death on long-bone length and symmetry, using whole-body micro-CT and semi-automatic length analysis, as we previously described[33]. At E17–P0, when cell death is past its peak, the left/right ratio of bone length reaches an average value of 0.77 for the femur, 0.90 for the humerus (Fig. 4a, b and Supplementary Fig. 3). The difference between forelimb and hindlimb is consistent with the fact that the *Pitx2-Cre* allele is more active in the prospective hindlimb region than the forelimb one[27]. Interestingly, by P2–3 the left/right femur ratio was significantly increased to 0.91 (Fig. 4a and Supplementary Fig. 3), indicating partial recovery of relative symmetry. This ratio plateaued at 0.9–0.95 by P5–P7, not showing further recovery by P14 (Fig. 4a), and keeping a similar value at P100, well past the end of the normal growth period (Fig. 4a and Supplementary Fig. 3). The results thus show that, as opposed to the prediction of the autonomous model, cell death in

the cartilage does trigger compensation, which can only be cell-nonautonomous (i.e. driven by the spared cells).

In growing limbs, the recovery of relative bilateral symmetry can have two causes: (1) a roughly constant absolute left-right (L-R) difference becomes relatively smaller as the animal gets bigger; (2) the generation of bone length at a faster rate in the injured limb than in the contralateral one, so that the absolute L−R difference becomes smaller with time. To determine which of these processes were participating in the observed recovery, we plotted the absolute L-R difference of bone length over time. These measurements showed that, in the first 2 weeks after birth, the L-R difference does not change significantly (Fig. 4b). Eventually, the absolute L-R difference did worsen (Fig. 4b, P100), but not proportionally to the increase in limb size, so that the L/R ratio did not change much as compared to P14 (Fig. 4a–c, Supplementary Fig. 3). Figure 4c showcases how a similar or bigger absolute L−R difference looks relatively minor at P14, as compared to being very evident at E17.5.

**Compensatory cartilage growth is driven by shifts in the proliferation-differentiation balance of chondrocyte progenitors, and by exacerbated cellular hypertrophy**
The results above suggest that, in response to chondrocyte loss, the spared cells exhibit compensatory behaviours that repair the damage to the cartilage and limit its impact on bone growth. First, to determine whether compensatory proliferation was taking place (akin to what we

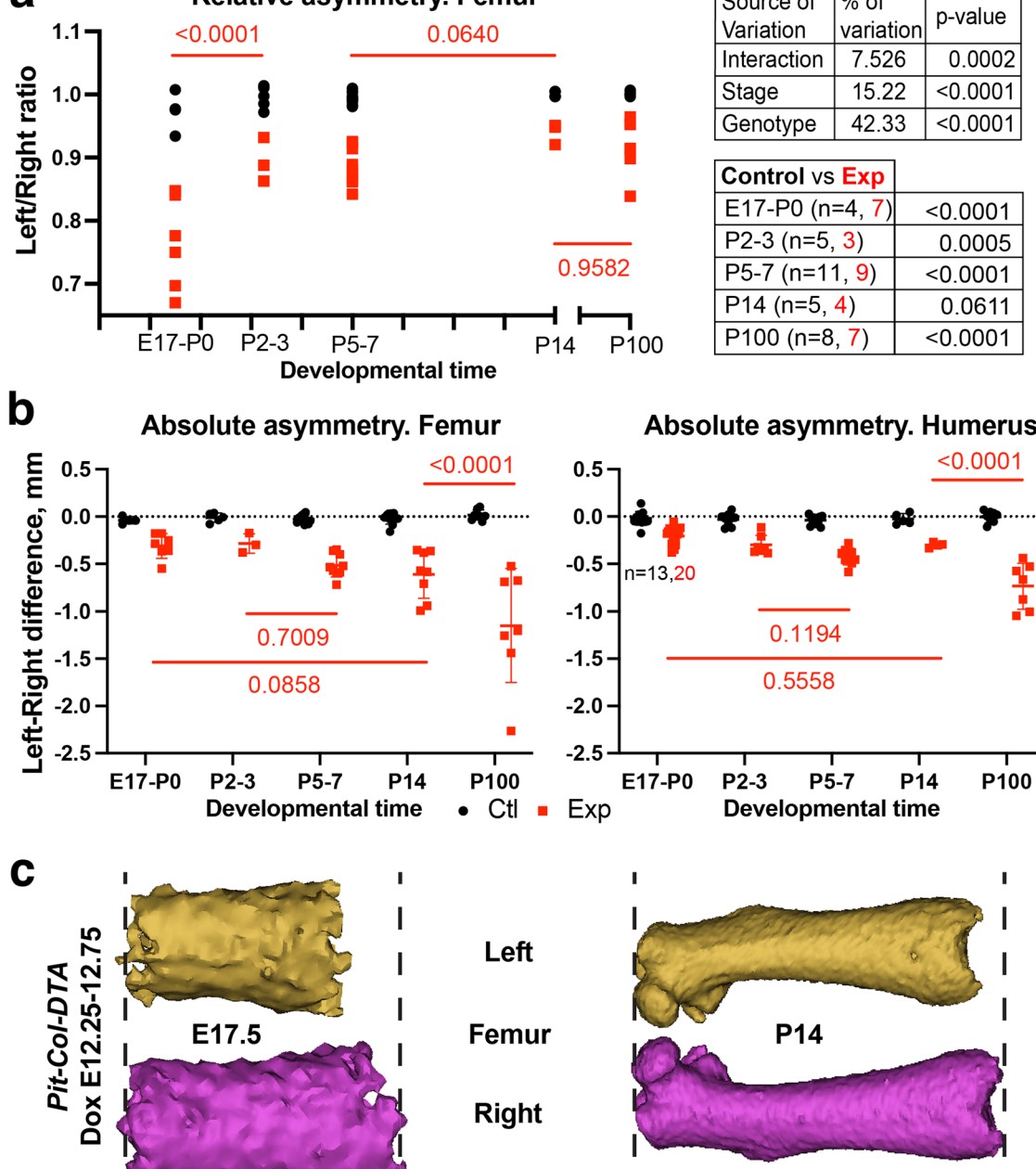

**Fig. 4 | After transient cell death, left-right limb asymmetry is followed by relative catch-up growth. a** Relative left-right difference (normalised over the right bone size) for control (black) and *Pit-Col-DTA* specimens (red) at the indicated stages. Values for 2-way ANOVA are shown in the Table on the top right. p-values of Sidak's multiple comparisons tests shown on the graph and the bottom right table. **b** Like **a**, but showing absolute asymmetry values. *n* as in (**a**), except where indicated. Lines = Mean ± SD. **c** 3D models of *Pit-Col-DTA* E17.5 and P14 experimental femora, reconstructed from micro-CT images. The models have been re-scaled to show a similar length of the shaft, to allow for comparison of the asymmetry at equivalent scales.

described in a mosaic model of cartilage cell-cycle arrest[27]), we quantified the incorporation of the thymidine analogue 5-Ethynyl-2-deoxyuridine (EdU), provided 1.5 h before collection. Interestingly, this assay did not reveal differences between Exp L and either Exp R or Ctl limbs at E17.5 or P0 (Supplementary Fig. 4), when the recovery process should be taking place.

We next reasoned that even if the instantaneous proliferation rate was not significantly changed, it was possible for the flux of resting to proliferative chondrocytes, and for the ratio of resting-chondrocyte self-renewal to change in response to injury (see predictions in Fig. 1b'–b"). To test this possibility, we performed pulse-chase experiments. We provided an EdU pulse at either E15.5, E17.5, P1, or

P3, and analysed the distribution of EdU+ cells two days later, i.e., at E17.5, P0, P3, or P5 (Fig. 5a). In the E15.5 → E17.5 experiment, we observed more EdU+ cells in the Exp L RZ as compared to Ctl, and fewer in the Exp L PZ as compared to Exp R (Fig. 5b). This result suggested the accumulation of cells in the RZ, which was confirmed by the progressive increase in RZ cell density observed from E15.5 to P0 Exp L cartilage (Supplementary Fig. 5a). To assess the speed of proliferation in the PZ, we quantified the number of columns with ≥4 EdU+ chondrocytes (C4+ hereafter), because two or more cell divisions were expected to take place in two days[34]. Fewer C4+ were found in the Exp L as compared to the Exp R and Ctl limbs (Fig. 5b'), confirming that both the generation and expansion of PZ cells slowed down. Overall,

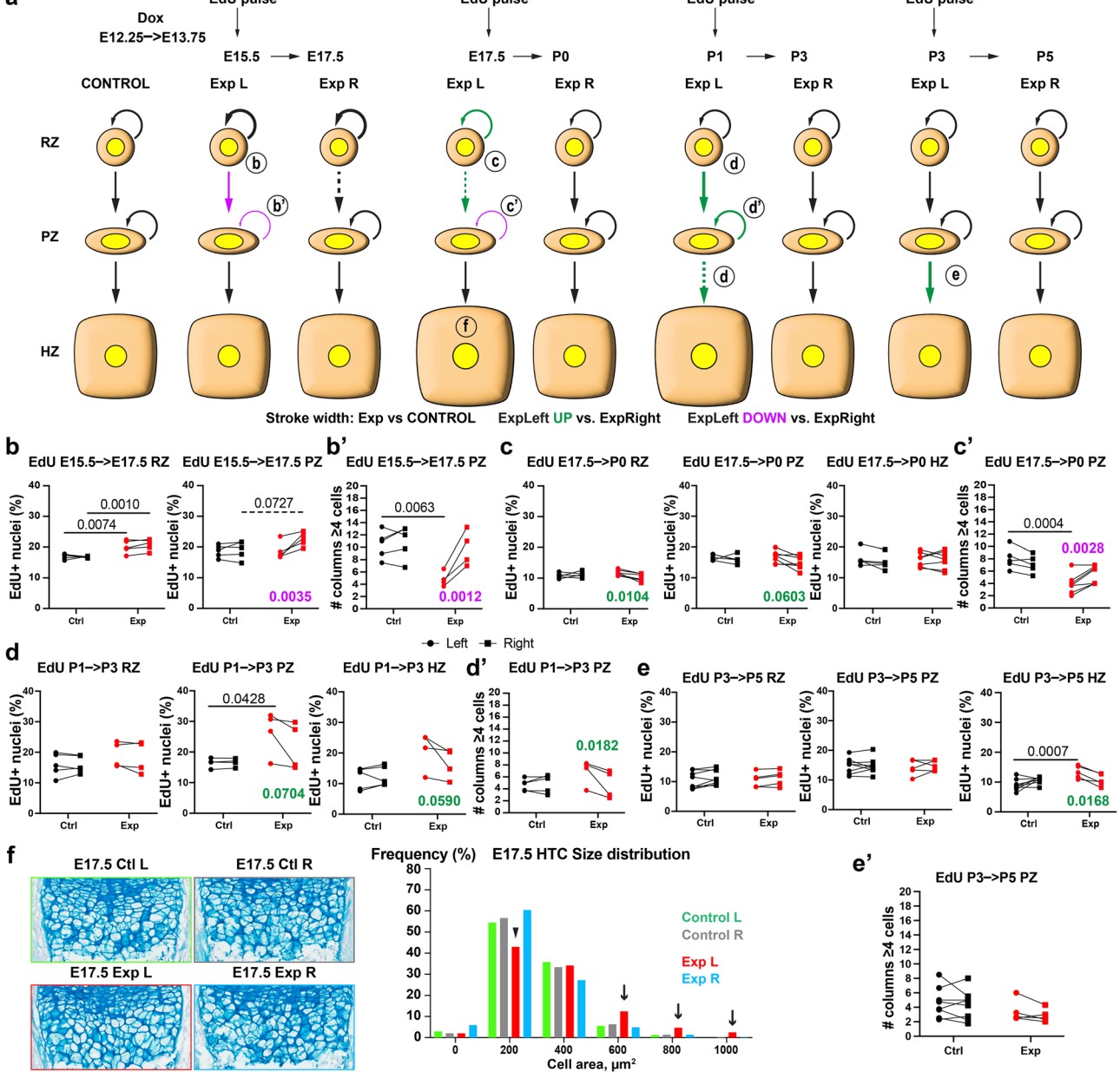

**Fig. 5 | Cartilage repair is driven by a proliferation-differentiation shift in chondrocyte progenitors, and by exacerbated cellular hypertrophy.**
**a** Schematic summary of the EdU pulse-chase experiments during and after transient unilateral cell death in the cartilage. The stroke width is used to compare Exp L vs. Ctl, whereas the colour is used to compare Exp L vs. Exp R, as indicated. The main changes are labelled with a circled letter (**b**–**e**), corresponding to the panel where the supporting evidence is presented. Dashed lines denote borderline significance. **b**–**e** Graphs showing the number of EdU⁺ nuclei (**b**–**e**) and the number of chondrocyte columns with ≥4 EdU⁺ cells (**b'**–**e'**), at E17.5 (**b**, $n = 5$ Ctl, 7 Exp), P0 (**c**, $n = 4, 4$), P3 (**d**, $n = 5, 4$) and P5 (**e**, $n = 8, 5$), in the indicated cartilage regions (RZ/PZ/HZ, resting/proliferative/hypertrophic). $p$-Values for post-hoc multiple comparisons tests after two-way ANOVA (Side × Genotype) are shown if <or close to 0.05. **f** Left: Cell size in the HZ is revealed by alcian blue staining of the extracellular matrix. Right: distribution of HTC sizes in the proximal tibia, for the indicated genotypes and sides, at E17.5 ($n = 3$ Ctl and 3 Exp). Genotypes and sides are colour-coded.

this early response, together with the cell loss due to cell death, likely explains why the height of the Exp L cartilage is reduced at E18.5 (Fig. 3e). Two days after an E17.5 pulse, the Exp L showed slightly increased number of EdU⁺ cells in the RZ (and a trend in PZ) as compared to Exp R, but not as compared to Ctl (Fig. 5c), and reduced number of C4+, as compared to the Exp R and Ctl (Fig. 5c'). These results suggest that the early response was starting to abate and even to be reversed. Indeed, in the P1 → P3 pulse-chase, EdU⁺ cells were more abundant in Exp L vs. Exp R across the PZ (Fig. 5d), and more C4+ were found in Exp L PZ (Fig. 5d'), confirming the inversion of the early response. This trend was then continued into the next cartilage layer,

as the P1 → P3 pulse-chase revealed increased flux towards the Exp L HZ (Fig. 5d), which was even more noticeable in the P3 → P5 experiment (Fig. 5e). In summary, in the first step of the response label-retaining progenitors were accumulated in the RZ, while in the postnatal response there was an accelerated transition towards the subsequent states.

It is noteworthy that, despite the decreased number of cells being produced during the injury phase, the length of the HZ was never much affected (Fig. 3e), suggesting the participation of other compensatory mechanisms. We thus measured HTC cell area on histological sections at multiple stages, finding that the size distribution in the

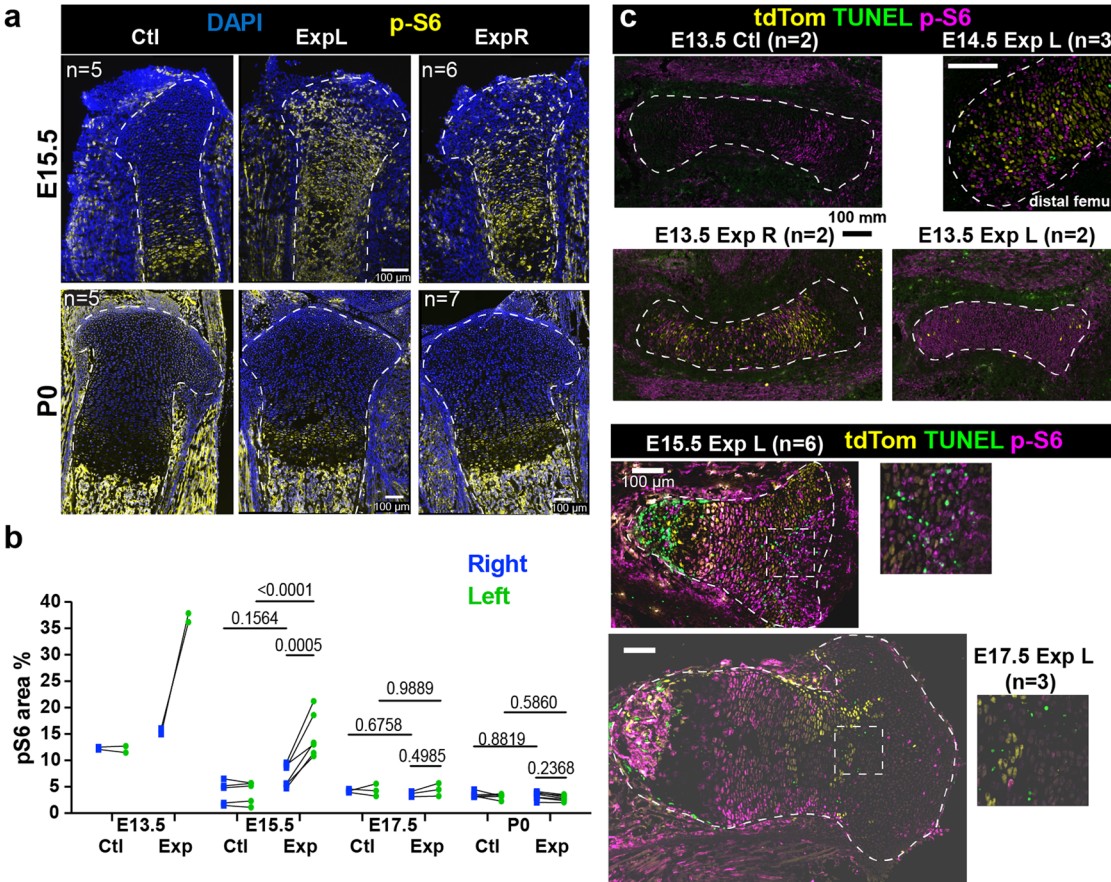

**Fig. 6 | Ectopic mTORC1 activity after cartilage cell death. a** p-S6 signal reveals mTORC1 activity in Ctl and Exp groups. Sample number (*n*) as indicated. **b** p-S6⁺ area, as % of the RZ + PZ (*n* = 2, 2, 5, 6, 3, 3, 5, 7, from left to right). **c** Time course of p-S6 immunostaining (dashed lines, cartilage), and close-ups to show the proximity with cell death (at E14.5 in the distal femur and E13.5 and E15.5 in the proximal tibia). n as indicated. Scale bars represent 100 μm.

Exp L HZ shifted away from small areas (Fig. 5f arrowhead) and towards high ones (Fig. 5f arrows), including values that were never found in Exp R or Ctl HZ. This compensatory hypertrophy only lasted for a few days, as the size distribution returned to being mostly normal by P3 (Supplementary Fig. 5b).

The retention of EdU⁺ cells in the RZ during the early phase of the recovery (Fig. 5b) resembled the enhanced self-renewal of Axin2⁺ chondroprogenitors observed during transient diet restriction in juvenile mice[28]. Since the capability of self-renewal in chondroprogenitors has been shown to appear only at postnatal stages[35], one possible explanation was that self-renewal had been precociously activated upon injury. The alternative scenario was that there was no enhanced production of RZ chondrocytes, but rather retention without proliferation. Importantly, the fact that the left/right ratio of instantaneous EdU incorporation increased in Exp animals between E15.5 and E17.5 (Supplementary Fig. 6) supported the first scenario, increased self-renewal. However, we were not able to formally test this, as the expression of the typical markers of <u>postnatal</u> long-term chondroprogenitors/cartilage stem cells such as Axin2, FoxA2 and PTHrP[36–38], was undetectable by immunofluorescence. Moreover, bulk RNA-seq of the growth plate did not reveal major changes in the otherwise low expression levels of the genes encoding these markers (Supplementary Fig. 7a, b).

### The mTORC1 pathway initiates the compensatory response in the injured cartilage

To uncover the molecular mechanisms that mediate the cellular behaviours described above, we performed bulk RNA-seq on left and right cartilage (from femur and tibia) of fetal and early postnatal mice (see Methods and Supplementary Fig. 7a). To reveal candidate genes and/or pathways involved both in the early response to injury and in the subsequent recovery, we performed Differential Gene Expression (DGE) analysis and Gene Set Enrichment Analysis (GSEA) between Exp L, Exp R and Ctl at E15.5, E17.5, P0 and P3 (Supplementary Fig. 7c-d and Supplementary Data 1–4). While several enriched pathways called our attention, assessment of candidate expression confirmed only a few of them as differentially expressed. Namely, we immuno-stained for phosphorylated ribosomal protein S6 (p-S6) as an mTORC1 activity readout, and found it ectopically expressed in the Exp L cartilage during the injury phase, going back to a normal state by P0 (Fig. 6a–c). Surprisingly, the Exp R cartilage also showed increased mTORC1 signalling as compared to Ctl cartilage at E15.5, although to a lesser extent than Exp L (see Discussion section). A time-course of p-S6 immunostaining revealed that p-S6 expression was broadly distributed in the E13.5 Exp L cartilage, (Fig. 6b, c), even though DTA expression was not so widespread (Supplementary Fig. 1b) and TUNEL signal only became detectable at E14.5 and in a patchy manner only (Figs. 2c and 6c). This suggests that the mosaic expression of DTA triggers a non-autonomous stress response in most of the cartilage, eventually leading to cell-autonomous death of some chondrocytes. At E15.5, the p-S6 signal was patchier, and closely associated with the dying cells (Fig. 6c insets), suggesting that cell death triggers a short-range response that maintains mTORC1 activity.

To determine the relevance of the ectopic expression of mTORC1, we set out to inhibit mTORC1 activity in vivo, during the prenatal period post-injury. *Pitx2-Cre; Col2a1-rtTA* females time-mated with

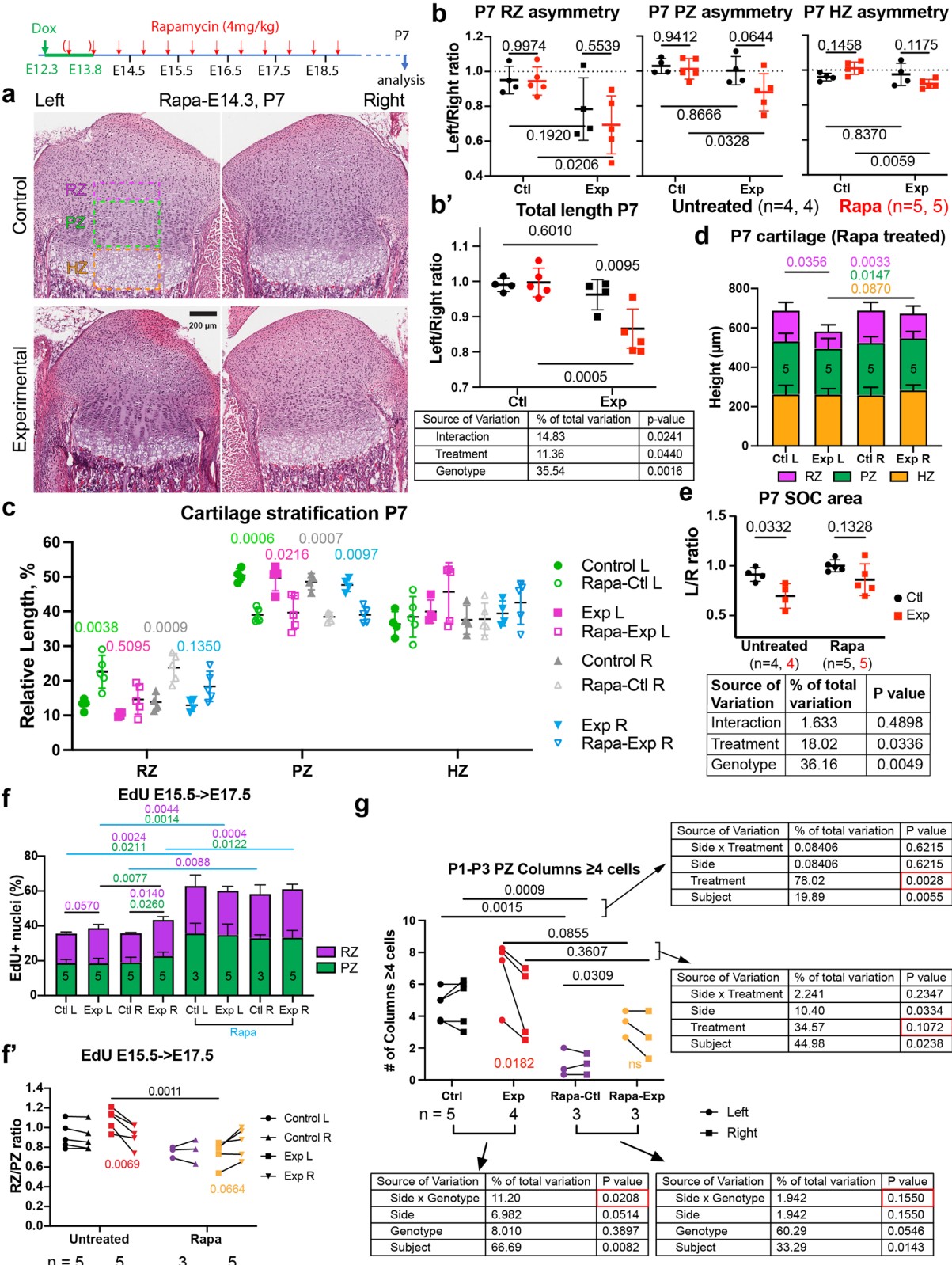

*Tigre*^*Dragon-DTA* males were provided with Dox from E12.25 to 13.75, and then with the mTORC1 inhibitor Rapamycin (4 mg/kg twice daily, Rapa hereafter) from E14.3 (or E13.3 in some cases) to E18.8 inclusive (Fig. 7a). The treatment did reduce ectopic mTORC1 activity, but did not completely abolish it, especially when initiated at E14.3 (Supplementary Fig. 8a). Importantly, cell death in the cartilage was not exacerbated by the treatment (Supplementary Fig. 8b), suggesting that

ectopic mTORC1 was not required to sustain survival of cells expressing borderline levels of our attenuated DTA version. Whereas Rapa treatment did not worsen the L/R asymmetry of experimental animals by P7 or P100 (Supplementary Fig. 8c, d), likely due to incomplete mTORC1 inhibition, it was enough to impair the reduction of acellular gaps by P7 (compare Supplementary Fig. 9a with Supplementary Fig. 2a) and to impede the subsequent recovery of cartilage

**Fig. 7 | mTORC1 signalling is required to repair the damaged cartilage.**
**a, b** Pregnant dams were treated with Rapamycin and pups collected at P7. H&E images (**a**) and the average of the longest length per section (**b, b′**) are shown as L/R ratios for both treatments. RZ/PZ/HZ, resting/proliferative/hypertrophic zones. Dotted lines: expected L/R ratio for controls. The table shows the results of the two-way ANOVA. *p*-Values for Sidak's multiple comparisons tests are shown on the graphs. Scale bars represent 200 μm. **c** Relative length of the different proximal tibia zones in P7 animals from untreated litters or treated *in utero* with Rapa. *p*-Values for Sidak's multiple comparisons test (after two-way ANOVAs) are shown. **d** Like **c**, for treated litters (# of samples shown inside the bars). **e** L/R ratio of the 2ry ossification centre area in untreated and Rapa treated P7 samples. Two-way ANOVA and Sidak's tests are shown. **f** Quantification of EdU pulse-chases for untreated and Rapa samples (RZ, PZ are colour-coded). *n* as indicated inside the green bars. After

two-way ANOVA (Side × Genotype for each Treatment separately, or Zone × Genotype for each Side separately) *p*-values for Sidak's multiple comparisons test were calculated, and significant ones (≤0.06) are shown colour coded. **f′** Quantification of RZ/PZ ratio of EdU+ cells for untreated and Rapa samples, Ctl and Exp (colour-coded). *n* as shown in f. After 2-way ANOVA (Side x Genotype for each Treatment separately, or Genotype × Treatment for each Side separately) *p*-values for Sidak's multiple comparisons test were calculated, and significant ones ( ≤ 0.06) are shown. **g** P1- > P3 EdU pulse-chase was used to quantify the number of PZ columns with >4 EdU+ cells. Two-way ANOVAS (Side × Genotype for the two treatments; Side × Treatment for the two Genotypes) and *p*-values for Sidak's multiple comparisons tests are shown. *n* = 5 Ctl, 4 Exp, 3 Rapa-Ctl, 3 Rapa-Exp. Mean and SD are shown in (**b–f**).

cytoarchitecture (Fig. 7b–d). Namely, in contrast to untreated litters, the left/right ratio of the length of each cartilage region in Rapa-treated litters was significantly smaller in experimental (rtTA+) animals at P7, as compared to controls (Fig. 7b, b′, red symbols). Interestingly, Rapa seemed to affect the control cartilage as well, especially the proliferative and resting zones. Consequently, the internal proportions (i.e. the stratification) of the control cartilage changed, with the RZ being significantly longer−and the PZ shorter−in treated animals as compared to untreated ones (Fig. 7c). While this trend was also observed in the Exp R cartilage, and presumably also happened in the Exp L, inhibition of the compensatory response by Rapa seemed to prevail in the Exp L cartilage. Indeed, the Exp L cartilage was shorter than the Exp R at P7, both at the RZ and PZ levels (Fig. 7d), whereas in untreated Exp animals all growth plate zones were similar between the left and right limbs (Fig. 3e). Along these lines, while the effect of Rapa on controls is towards increasing the absolute length of the RZ (Supplementary Fig. 9b), in Exp L this effect is masked by the failure of the compensatory response when mTORC1 is inhibited. Lastly, we measured the SOC in the P7 Rapa-treated samples, and found that the Exp L/R ratio was not significantly different from that of Control animals (Fig. 7e), unlike in untreated samples (see Supplementary Fig. 2b, c). These results indicated that the delay in Exp cartilage progression also requires mTORC1 activity.

In terms of mechanism, mTORC1 inhibition did not impair the exacerbated hypertrophy of Exp L HTCs at E17.5 (Supplementary Fig. 9c), suggesting that the main defect was at the cell-number level. Surprisingly, Rapa did not affect HTC size in control cartilage either (Supplementary Fig. 9c), in contrast with the reported effects of the genetic loss-of-function of mTORC1 signalling[39–42].

To test potential mTORC1 effects at the cell-number level, we repeated the EdU pulse-chase experiments in Rapa litters. Of note, Rapa treatment led to an overall increase in the number of EdU+ cells in RZ and PZ of both Exp and Ctl samples (Fig. 7f, blue lines). Thus, to facilitate comparisons, we also calculated a RZ/PZ ratio for every sample (Fig. 7f′). As mentioned before (Fig. 5), in untreated samples exposed to an E15.5 → E17.5 pulse-chase, we observed decreased number of EdU+ PZ cells in Exp L vs. Exp R (Fig. 7f green *p*-values), as well as increased RZ/PZ ratio in Exp L vs. Exp R (Fig. 7f′ red *p*-value). Conversely, in Rapa litters, the number of EdU+ PZ cells was not different in Exp L vs. Exp R (Fig. 7f) and the RZ/PZ ratio was reduced (instead of increased) in Exp L vs. Exp R (Fig. 7f′, ochre *p*-value). Importantly, this happened without changes in proliferation rate, measured by short-term EdU incorporation (Supplementary Fig. 8e). These results suggest that, during the early stages of catch-up growth, mTORC1 is mostly involved in promoting the self-renewal behaviour of the RZ, at the expense of the transition towards PZ. Moreover, we analysed the number of C4+ in the PZ of P1 → P3 pulse-chased samples, and found that this parameter was dramatically reduced in Rapa vs. Untreated Ctl animals (Fig. 7g, purple), whereas the extreme Left-Right difference in Exp Untreated samples (Fig. 7g, red) became non-significant in Rapa samples (Fig. 7g, ochre). Therefore, mTORC1 is also

required to promote extra rounds of PZ cell proliferation during the late stages of catch-up growth, as well as in control conditions.

## Discussion

### Catch-up growth as a tool to uncover the mechanisms controlling organ size and repair

Developmental robustness is a biological phenomenon that can not only inspire regenerative therapies but also reveal mechanisms that regulate organ growth. In this study we set out to explore the molecular and cellular mechanisms behind catch-up growth of the long bones. This is a unique organ model that grows via a cartilage template that is produced and destroyed in opposite sites at the same time. This process requires tight control of the proliferation and differentiation of cartilage progenitors, and leads to the different chondrocyte layers being organised in a distinct cytoarchitecture. The proportions of this layered structure often change across different growth plates, developmental stages and species, and have been shown to correlate with the speed and/or extent of long bone growth[43,44]. However, how these proportions are established and/or recovered after a perturbation is not completely understood, and was explored in this study.

### A mouse model for acute cell death in the growth plate cartilage

We capitalised on the *Dragon-DTA* allele that we described previously[30] to develop a model of acute unilateral injury in the cartilage. A brief pulse of Dox was enough to induce transient expression of DTA and subsequent cell death in the fetal cartilage, mostly in the left limb (Fig. 2). Unsurprisingly, the extensive cell death that takes place in the Exp L cartilage caused major disruptions in its integrity, with acellular gaps that in some cases extended across almost the whole growth plate (Fig. 3). As a consequence, the length of the cartilage growth plate was altered, especially in the case of the RZ and PZ. Given that the growth plate is the template for bone growth, this model also showed a remarkable growth impairment of the left-limb bones, causing a peak left-right difference of 25-30% by P0 (Fig. 4). At this point cell death was over, potentially allowing for recovery to happen. This model thus provides at the same time exquisite tissue specificity and control of the timing and duration of the injury. It is also versatile, as it can be combined with any Cre and (r)tTA lines to affect virtually every tissue of interest, which could be very valuable for developmental and regenerative biologists.

### Recovery of cartilage integrity and cytoarchitecture after cell death is in part mediated by mTORC1 activity

After the injury, we observed a remarkable recovery of cartilage integrity and cytoarchitecture (Fig. 3), associated with a biphasic cellular response (Fig. 5): first, increased retention of EdU+ chondrocytes in the resting zone between E15.5 and E17.5, at the expense of the transition to the proliferative pool; second, a reversal of this balance, i.e. accelerated transition towards first the proliferative and then the hypertrophic pools between P1 and P5. During the first phase, compensatory hypertrophy also took place in the HZ, explaining why its

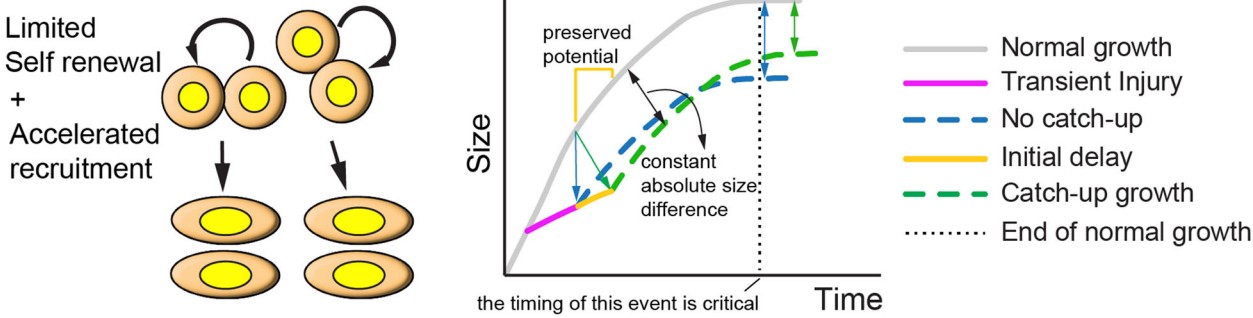

**Fig. 8 | A hybrid model of catch-up growth, in which a delay phase after the injury freezes up the growth potential, so that when growth resumes, it does it at an intermediate speed.** Normal size is never recovered, but the deficiency does not become worse. Note the improvement compared to the absence of catch-up.

size was not affected despite the reduced cell output from the upstream layers. In summary, cell-nonautonomous mechanisms were triggered by cell death, suggesting a key role of cell-cell communication in the response.

One of the cellular communication pathways that has been shown to impact bone growth is the insulin-like growth factor (IGF) pathway. Most of the IGF1/2 that act on the growth plate are produced locally[45–47], but their effects are region-specific. In resting cells, IGF1 promotes the recruitment towards proliferative chondrocytes, for example by repressing expression of PTHrP[48]. In addition, Oichi et al. showed that, during dietary restriction in juvenile mice, $Axin2^+$ progenitors in the cartilage were biased towards self-renewal instead of transitioning to the proliferative pool, which associated with decreased expression of $Igf1$ in the resting zone[28]. Conversely, this bias and the reduced expression of $Igf1$ were reversed when the food supply was restored, leading to catch-up growth[28]. IGF1 also plays a key role in the cartilage HZ, where it was shown to promote chondrocyte enlargement, alone or via interaction with other ligands[49–51], and to mediate the differential enlargement exhibited by HTCs located in different growth plates of the same animal[51].

One of the usual downstream effectors of IGF signalling with reported effects on resting and hypertrophic chondrocytes is the mechanistic target of rapamycin (mTOR, reviewed in[52]). Specifically, it has been shown that mTOR complex 1 (mTORC1) promotes bone growth via stimulation of protein synthesis and cell size in the HZ[39–42]. On the other hand, postnatal activation of mTORC1 in the RZ associates with a switch to self-renewal and increased clonogenic capacity of chondrocyte progenitors[35]. In this regard, mTORC1 was shown to enhance expression of PTHrP[53], a marker of so-called cartilage stem cells[36]. In summary, since mTORC1 regulates growth-related processes across multiple regions of the growth plate, it stands as a main candidate to coordinate the relative height of growth plate zones, and associated bone growth parameters. Based on this framework and our RNA-seq data, we hypothesised that the compensatory behaviours that we observed could be associated with increased mTORC1 activity, which was confirmed at the histological level by assessing expression of p-S6 (Fig. 6). Interestingly, p-S6 has been recently associated with the injury-associated area in different types of skin wounds[29], muscle injuries[54] and limb amputation[55], suggesting that it is a conserved response. In the case of skin wounds, p-S6 was shown to be a modulator of the response, but not strictly necessary for the response to happen[29]. It remains to be determined whether this is also the case in our model of catch-up growth, as we could not achieve full in vivo inhibition. Partial inhibition did however impair the recovery of cartilage cytoarchitecture in our model (Fig. 7), suggesting it is quite important. A priori, this role of mTORC1 could seem at odds with the findings by Oichi et al.[28], as in their model of catch-up growth they showed that the bias towards self-renewal at the expense of transition to the proliferative pool was associated with decreased expression of

$Igf1$ in the resting zone, which could presumably result in reduced mTORC1 signalling. However, mTORC1 activity was not assessed in their study, leaving open the possibility for it to be active, and thus independent of IGF1 in the RZ. Alternatively, mTORC1 may only be required in the response to physical damage (our study), but not in diet restriction (Oichi's study), or only during fetal stages (our study) and not in juvenile mice (Oichi's study). Additional studies will help elucidate these questions.

## A hybrid model of catch-up growth

As a consequence of impairing Exp L cartilage function and integrity, our unilateral injury model also caused a remarkable asymmetry in bone length, with the left femur being ~25% shorter by E17.5-P0 (Fig. 4). In terms of catch-up growth, while the relative asymmetry became milder with time, suggesting functional recovery, the absolute left-right difference did not improve over the whole growth period. In order to classify this behaviour as catch-up growth or not, it is important to consider that, in a growing animal, triggering cell death in the cartilage also ablates the long-lived chondroprogenitors. Therefore, this is expected to not only reduce the amount of new bone produced during cell death (and shortly after), but also to impair the subsequent growth potential, due to reduction of the pool of progenitors. In summary, in the absence of compensatory mechanisms, the absolute left-right difference is expected to continuously grow over time (Figs. 1b' and 8). On the other side of the spectrum, if there are compensatory mechanisms that can fully replenish the pool of progenitors, so that there is no loss of potential despite a transient stalling, then absolute asymmetry should decrease over time, as long as there is enough time to complete the growth period (Fig. 1b"). In the former case, the growth curve shifts vertically, whereas in the latter, the growth curve shifts horizontally. A novelty of this study is that our results suggest a hybrid model, in which part of the potential is lost during the injury, but the remaining potential is preserved during the stalling phase, such that when growth resumes, the growth curve shifts diagonally (Fig. 8). Interestingly, this would lead to the absolute left-right difference being roughly maintained over the linear growth period, which is what we observed (Fig. 4). Of note, the preservation of growth potential that we postulate for this model would be the first case in which this behaviour has been observed during fetal stages, before the radical switch in clonal behaviour that happens postnatally[35]. It remains to be determined whether all chondroprogenitors are capable of this behaviour, or just a sub-population of them.

Importantly, to maximize the catch-up in the models depicted in Fig. 1b" and 8, it is necessary that the growth period is extended beyond the normal time (dotted line in Fig. 8). This is why we analysed our model at P100, instead of at 9 weeks of age, (i.e., when normal longitudinal growth ceases in mice). It follows that if there is a systemic growth-halt before full local recovery has happened, catch-up growth

will be interrupted, which is something we cannot discard in our study. Future studies will need to investigate whether this is the case and, if so, how to extend the growth period beyond the normal time, to improve the efficacy of growth therapies.

## The control of normal cartilage proportions by mTORC1

Besides the role of mTORC1 in the compensatory response, our study has also revealed a potential role in the maintenance of the layered structure of the cartilage during normal growth. Indeed, treatment of control animals with Rapamycin led to changes in the relative proportions of RZ, PZ and HZ (Fig. 7). At the cellular level, we showed that mTORC1 inhibition caused increased retention of EdU+ cells between E15.5 and E17.5 in both RZ and PZ, suggesting an overall deceleration of cartilage progression. This was followed by a massive drop in the potential of proliferative chondrocytes to undergo subsequent rounds of cell division between P1 and P3, potentially explaining the reduction of the relative PZ proportion by P7. Given how variable cartilage stratification is across developmental stages, different bones and different species, further exploring the role of mTORC1 in coordinating the size of the growth plate zones is warranted. At the molecular level, we speculate that it could be mediated via its known interactions with IHH[56] and PTHrP[53], and other interactions yet to be determined.

## A potential systemic response triggered by the unilateral injury

During characterisation of the injury model, we realised that mTORC1 activity was also ectopically activated in the Exp R cartilage, especially after E15.5 (Fig. 6). While at first this may seem like a direct effect of sublethal DTA expression in the right cartilage, or just an effect of the genetic background (i.e. rtTA+ vs. rtTA−) several lines of evidence reject these possibilities. First, the number of DTA+ cells detected by in situ hybridization in the Exp R cartilage is not as abundant or widespread as the ectopic p-S6 signal (Supplementary Fig. 1). Second, we could not detect ectopic cell death or p-S6 expression in the Exp R cartilage in the absence of Dox (Supplementary Fig. 10a), indicating that the genetic background per se was not responsible for the cross-limb activation. Third, ectopic p-S6 was also found in the E15.5 left and right experimental rib cartilage, as compared to controls, even in the absence of ectopic cell death (Supplementary Fig. 10b, b'''). Lastly, the timing and pattern of the activation of mTORC1 are quite different between Exp L and Exp R cartilage. Indeed, while p-S6 was detected all over the Exp L cartilage as early as E13.5 (Fig. 6), its expression in the Exp R cartilage was not obvious until E15.5, once expression in Exp L had already switched to a salt & pepper pattern. These results suggest that there are two waves of ectopic mTORC1 activity in the Exp L cartilage, and that the Exp R only experiences the second one. We hypothesise that the cartilage injury is triggering not only a local response but also a systemic one, which includes activation of mTORC1 at distant sites. There are several examples in the literature where this has been observed. For example, unilateral injury in the limb muscles in adult mice leads to activation of mTORC1 signalling and of the so-called G$_0$ alert state in muscle stem cells of not only the injured limb, but of the uninjured one as well[54]. Similarly, unilateral limb amputation in axolotl leads to systemic activation of stem cells, including increased mTORC1 signalling and additional rounds of cell replication[55]. Interestingly, in our model we also detected cell behaviour changes in the Exp R chondrocytes, such as the increase retention of EdU+ chondrocytes in the resting zone between E15.5 and E17.5, despite the absence of a major injury (Fig. 5). Importantly, we would not have been able to detect this contralateral effect with traditional approaches, as fine manipulation of a specific limb tissue is nearly impossible in fetal mice, while traditional genetic injury models typically affect left and right limbs equally. From an evolutionary perspective, this type of systemic priming could prepare other regions of the body to respond faster, providing adaptive value in the case that a second injury takes place. Indeed, in the two studies mentioned above,

the primed cells were able to respond faster to secondary injuries[54,55]. While outside the scope of this study, it would be interesting for future studies to determine whether distant priming prepares the other limb to respond better to injury in our model of compensatory growth, and whether this response extends to other types of tissues.

# Methods

## Animal models

The use of animals in this study was approved by the Monash Animal Research Platform animal ethics committee at Monash University (protocol #32263). The *Pitx2-ASE-Cre* (aka *Pitx2-Cre*) mouse line, obtained from Prof. Hamada[31], was crossed with the *Col2a1-rtTA* mouse line[32], and then inter-crossed to generate *Pitx2-Cre/Cre; Col2a1-rtTA* mice. Genotyping was performed as described previously[27]. The *Tigre^Dragon-DTA* mouse line was described in[30]. Experimental and control animals were generated by crossing *Pitx2-Cre/Cre; Col2a1-rtTA/+* females with *Tigre^Dragon-DTA/DTA* males (i.e. homozygous for the conditional misexpression allele). The separation of control and experimental animals was based on the rtTA genotype. Pregnant females were administered doxycycline hyclate (Sigma, 1 mg/ml in drinking water, with 0.5% sucrose for palatability) from E12.25 to E13.75. The day of vaginal plug detection was designated as E0.5, and E19.5 was referred to as P0. All animals used for mating were aged between 8-weeks to 9-months. These animals were housed in special cage with 12 h dark and 12 h light cycle, ambient temperature and humidity.

## Drug administration

A solution of Rapamycin (Selleckchem #S1039) was prepared at the concentration of 2 mg/ml in a mixture of water, 2% dimethylsulfoxide (DMSO), 30% polyethylene glycol 300 (PEG 300), and 5% polysorbate 80 (Tween 80). This solution was administered to the pregnant female via intraperitoneal (i.p.) injection at a dose of 4 mg/kg (twice daily).

## EdU incorporation and detection

A solution of 5-Ethynyl-2'-deoxyuridine (EdU) was prepared at 6 mg/ml in phosphate-buffered saline (PBS). This solution was administered at a dose of 30 μg/g, subcutaneously (s.c) for pups and intraperitoneally (i.p.) for pregnant females, 1.5 h before euthanizing the mice (or 2 days before, for pulse-chase experiments). To detect EdU, a click chemistry reaction with fluorescein-conjugated azide (Lumiprobe #A4130) was performed once the immunohistochemistry and/or TUNEL staining were completed on the same slides. Briefly, the working solution was prepared in PBS, adding CuSO$_4$ (Sigma # C1297) to 4 mM, the azide to 0.4 μM and freshly-dissolved ascorbic acid (Sigma #A0278) to 20 mg/ml, and incubating the sections for 15 min at room temperature in the dark, followed by PBS washes.

## Sample collection and processing

Mouse embryos were euthanized using hypothermia in cold PBS, while mouse pups were euthanized by decapitation. Upon collection of the embryos or pups, the limbs (including full tibiae and/or femora) were carefully dissected out in cold PBS, skinned, and fixed in 4% paraformaldehyde (PFA) for 2 days at 4 °C. Samples of P1 or younger were not subjected to decalcification. For P3, P5, and P7 samples, decalcification was performed by immersing the specimens in 0.45 M ethylenediaminetetraacetic acid (EDTA) in PBS for 3, 5, and 7 days, respectively, at 4 °C. Following several washes with PBS, the limb tissues were cryoprotected in PBS containing 15% sucrose and then equilibrated in 30% sucrose at 4 °C until they sank. The hindlimbs were then oriented sagittally, facing each other, with the tibiae positioned at the bottom of the block (closest to the blade during sectioning) and embedded in Optimal Cutting Temperature (OCT) compound using cryomolds (Tissue-Tek). The specimens were frozen by immersing them in dry-ice-cold iso-pentane (Sigma). Serial sections with a thickness of 7 μm were collected using a Leica Cryostat on Superfrost slides.

The sections were allowed to dry for at least 30 min and stored at −80 °C until further use. Prior to conducting histological techniques, the frozen slides were brought to room temperature in PBS, and the OCT compound was washed away with additional rounds of PBS rinses.

## Micro-CT and bone length analysis

The micro-CT and bone length analysis was as previously described[33]. Briefly, samples were retained and fixed in 4% PFA as residual tissues from other experiments in the Rosello-Diez lab. Whole femora and humeri were scanned using a Siemens Inveon PET-SPECT-CT small animal scanner in CT modality (Monash Biomedical Imaging). The scanning parameters included a resolution of 20 and 40 μm, 360 projections at 80 kV, 500 μA, 600 ms exposure with a 500 ms settling time between projections. Binning was applied to adjust the resolution with 2 × 2 for 20 μm scans and 4 × 4 for 40 μm scans. The acquired data were reconstructed using a Feldkamp algorithm and converted to DICOM files using Siemens software. For the analysis and bone length measurements, Mimics Research software (v21.0; Materialize, Leuven, Belgium) equipped with the scripting module was utilized to develop the analysis pipeline.

## Histology, haematoxylin and eosin and alcian blue staining

Following sectioning, the sections were washed in distilled water. For Alcian blue staining, the sections were incubated in a 1% Alcian Blue solution in distilled water, adjusted to pH 1.0 with hydrochloric acid (Sigma #A5268), for 15 min at room temperature, followed by rinsing in water. For haematoxylin and eosin (H&E) staining, the sections were stained with filtered mercury-free Harris haematoxylin solution (Point of Care Diagnostics #VWRC351945S) for 5 min, washed in running tap water for 5 min, and then rinsed in 95% ethanol. Next, the sections were counterstained with a 0.25% Eosin Y solution (Sigma #HT110116) in 80% ethanol and 0.5% glacial acetic acid for 1 min. For both Alcian Blue and H&E staining, dehydration was performed by passing the sections through 95% ethanol, absolute ethanol, and xylene. Finally, the sections were mounted using a xylene-based mounting medium, dibutylphthalane polystyrene xylene (DPX) (Sigma #100579).

## Immunohistochemistry and TUNEL staining

For antigen retrieval, the sections were subjected to citrate buffer (10 mM citric acid, 0.05% Tween 20 [pH 6.0]) at 90 ˚C for 15 minutes. Afterward, the sections were cooled down in an ice water bath, washed with PBSTx (PBS containing 0.1% Triton X-100). To perform TUNEL staining, the endogenous biotin was blocked using the Avidin/Biotin blocking kit (Vector #SP-2001) after antigen retrieval. Subsequently, TdT enzyme and Biotin-16-dUTP (Sigma #3333566001 and #11093070910) were used according to the manufacturer's instructions. Biotin-tagged DNA nicks were visualized using Alexa488- or Alexa647-conjugated streptavidin (Molecular Probes #S32354 and #S32357, diluted 1/1000) during the incubation with the secondary antibody.

For immunohistochemistry staining, sections were incubated with the primary antibodies prepared in PBS for either 1.5 h at room temperature or overnight at 4 °C (see list of antibodies below). Following PBSTx washes, the sections were incubated with Alexa488-, Alexa555-, and/or Alexa647-conjugated secondary antibodies (ThermoFisher #A21206, #A21432, #A31573; diluted 1/500 in PBSTx with DAPI) for 1 hour at room temperature. After additional PBSTx washes, the slides were mounted using Fluoromount™ Aqueous Mounting Medium (Sigma). The antibodies used, along with their host species, vendors, catalogue numbers, and dilutions, were as follows: mCherry (goat polyclonal, Origene Technologies #AB0040-200, diluted 1/500), p-S6 (rabbit polyclonal, Cell Signaling Technology #2211S, diluted 1/300).

## In situ hybridization

To obtain the DTA riboprobe, the coding sequence of DTA was first amplified from the pB6-Rosa26-DTA-Soriano-for vector using primers XhoI_K_DTA_F (gactgacctcgaggccaccATGGAAGCGGGTAGGCCTT) and ClaI_DTA_R (gactgacatcgatTTAGAGCTTTAAATCTCTGTAGGTAGTT), and directionally cloned into XhoI-ClaI digested pBS KS plasmid. The resulting plasmid was linearized with XhoI and purified by phenol/chloroform extraction. In vitro transcription was performed on the linearized template using T7 RNA polymerase (Roche DIG-labelling mix) to obtain a full-length Digoxigenin-labelled *DTA* riboprobe, which was purified via the LiCl/EtOH precipitation method, and resuspended in RNAse-free water to a concentration of ~400 ng/μl. The pB6-ROSA26-DTA-Soriano-for vector was a gift from Mario Capecchi (Addgene plasmid #125745; http://n2t.net/addgene:125745; RRID:Addgene_125745).

To perform in situ hybridisation, sections were fixed in 4% PFA for 20 minutes at room temperature, washed in PBS, and treated with 4 μg/ml Proteinase K for 15 min at 37 °C. After washing in PBS, the sections were refixed with 4% PFA, followed by treatment with 0.1 N pH8 triethanolamine (Sigma #90279), 0.25% acetic anhydride (Sigma #320102) for 10 minutes at room temperature. Subsequently, the sections were washed in PBS and water and incubated with prehybridization buffer (50% formamide, 5× SSC pH 5.5, 0.1% 3-[(3-Cholamidopropyl)dimethylammonio]−1-propanesulfonate (CHAPS), 0.05 mg/ml yeast tRNA, 0.1% Tween 20, 1× Denhardt's) at 60 °C for 30 min. The sections were then incubated with 1 μg/ml preheated riboprobes and subjected to hybridization at 60 °C for 2 h. Post-hybridization washes were performed using post-hybridisation buffer I (50% formamide, 5x pH 5.5 SSC, 1% SDS) and II (50% formamide, 2× pH 5.5 SSC, 0.2% SDS) preheated at 60 °C for 30 min, respectively. The sections were then washed with maleic acid buffer (MABT: 100 mM maleic acid, 150 mM NaCl, 70 mM NaOH, 0.1% Tween 20) and blocked with 10% goat serum, 1% blocking reagent (Roche #11096176001) in MABT at room temperature for 30 minutes. Next, the sections were incubated overnight at 4 °C with anti-digoxigenin-AP (Sigma #11093274910) diluted 1/4000 in MABT with 2% goat serum and 1% blocking reagent. After several MABT washes, the sections underwent incubation with AP buffer (0.1 M Tris-HCL pH 9.5, 0.1 M NaCl, 0.05 M MgCl$_2$, 0.1% Tween 20), with the second one containing 1 mM levamisole and then developed colour using BM purple (Roche #11442074001) at 37 °C. Following a wash in PBS, the sections were fixed for 10 min in 4% PFA, counterstained with Nuclear Fast Red (Sigma #N3020) at room temperature for 10 min, and rinsed in water. Dehydration was performed by passing the sections through 70%, 90% ethanol, absolute ethanol, and xylene. Finally, the sections were mounted with DPX (Sigma #100579).

## Imaging

Sagittal sections of the limbs were captured, with a focus on the area between the distal femora and proximal tibiae. Typically, at least 2 sections per limb were analyzed, although in most cases 4 sections were examined. In the case of cultured distal femora, frontal sections were used as they provided better identification of the different epiphyseal regions. To determine the boundaries of the resting zone (RZ), proliferative zone (PZ) and hypertrophic zone (HZ), morphological criteria were applied. The transition between round (resting) and flat (columnar) nuclei, forming an arch along the upper point of the grooves of Ranvier, was considered as the start of the PZ. On the other hand, the transition towards larger, more spaced nuclei (pre-hypertrophic) marked the end of the PZ. The point where the pericellular matrix exhibited a sharp reduction around enlarging chondrocytes was designated as the beginning of the HZ. The distal end of the last intact chondrocyte served as the endpoint of the HZ. For imaging, bright-field and fluorescence images were acquired using a Zeiss upright microscope (Imager.Z1 or Z2) equipped with Axiovision software (Zeiss/ZenBlue). Mosaic pictures were automatically generated by assembling individual tiles captured at 10× magnification for bright-field images or 20× magnification for fluorescence images.

## Image analysis and quantification

The regions of interest such as resting, proliferative and hypertrophic zones (RZ, PZ, HZ) and the secondary ossification centre (SOC) were identified from imaged sections of the multiple models (left and right; experimental and control proximal tibial cartilage). Consistent parameters such as brightness, contrast, filters and layers were kept the same for all the images in the same study.

## Cell count analysis

The RZ and PZ were identified and segmented from sections stained for DAPI, tdTomato, and TUNEL or EdU using FIJI software. The number of cells in the region of interest (RZ & PZ) was measured using Cell Profiler. In rare cases, outliers or technically challenging images were discarded from analysis (red numbers in raw data file). DTA$^+$ cells were quantified manually on H&E images using FIJI software (Multi-point tool).

## Cell size analysis

The HZ was identified and segmented from sections stained for Alcian Blue. The area ($\mu m^2$) of individual hypertrophic chondrocyte was measured using FIJI software. Data from at least 3 mice (6 limbs), and 2-4 sections per limb, was pooled together to build the histograms in GraphPad Prism.

## Cartilage length analysis

Using H&E staining, distinct zones of cartilage were identified. The length ($\mu m$) of each cartilage zone (RZ, PZ, HZ) was then measured using FIJI software.

## p-S6 area percentage analysis

Following DAPI and p-S6 staining, the area percentage (%) of p-S6 signals over the total area of region of interest (RZ + PZ) were measure based on the signal threshold using FIJI software.

## Secondary ossification centre (SOC) relative area analysis

Following H&E staining, the SOC area (including blood vessel invasion areas and hypertrophic cells) and the total epiphyseal area were measured using FIJI software.

## Statistical analysis

Statistical comparisons were performed using appropriate tests based on the experimental design. An unpaired t test was used for comparisons involving one variable and two conditions. Two-way ANOVA was utilized for comparisons involving two variables and two or more conditions. Non-parametric tests were selected when the assumption of normality could not be met. All statistical analyses were conducted using Prism9 software (GraphPad).

## Multiplexed RNA-seq and analysis

**Experimental Design.** The RNA-Seq study has a 2 (genotype: control, experimental) by 2 (side: left, right) by 4 (stage: E15.5, E17.5, P0, P3) factorial design. We ensured a minimum number of 3 biological replicates per experimental group were sequenced (2 for controls). The final experimental design is summarised in Supplementary Fig. 7. For the stages P0 and P3 there were two technical replicates per biological sample, which were subsequently merged after confirming their similarity.

**RNA extraction, multiplexed library preparation, sequencing.** Control and experimental samples (2-3 biological replicates each) were collected for 4 different stages: E15.5, E17.5, P0 and P3. Left and right cartilage samples (each containing proximal and distal tibia and femur) were kept separated for each sample, and flash-frozen upon dissection. Total RNA was extracted using the Monarch Total RNA Miniprep Kit (New England Biolabs), following manufacturer instructions. From this, RNA-Seq of messenger RNAs (mRNAs) was performed using a custom in-house multiplex method, which allowed us to sequence up to 24 different samples in the same sequencing lane. Briefly, samples were given a unique i7 index (together with UMI) during individual polyA-priming and first-strand synthesis, which also added a template-switch sequence to the 5' end. Samples were then pooled and amplified using P7 and an oligo which binds the template-switch sequence. Final library construction was completed by tagmentation and addition of P5 (with i5 index) by PCR. Sequencing was performed on an Illumina NSQ2k run with up to 101nt SR (cDNA). An 18-nt i7 read contains the 8-nt index and 10-nt UMI and, where required, an 8-nt i5 index read is also generated. We obtained a read depth of at least 15 million reads per library.

**RNA-Seq data analysis.** Raw data was examined by FASTQC and MULTIQC for read quality, adapter contaminations and presence of overrepresented sequences[57]. i7 index reads were trimmed (the first 8 bp removed) and remaining 10-bp UMI was identified per sample using cutadapt[58]. The tools umi-tools[59] (v 0.5.5) and Bbmap[60] (v. 38.81) were used to demultiplex the samples based on the identified barcodes. Next, STAR aligner[61] (v 2.7.9a) was used to map the demultiplexed fastq reads to the mm39 (GRCm39) reference genome from Ensembl with the following parameters:--outFilterMultimapNmax 20, --outFilterMismatchNmax 5, --outFilterScoreMinOverLread 0. featureCounts() function from the R/Bioconductor package Rsubread[62] (v 2.10.4) was used to generate the gene count matrix for the aligned reads with following parameters; allowMultiOverlap = TRUE, minOverlap = 1, fracOverlap = 0, fracOverlapFeature = 0. LIMMA (Linear Models for Microarray Data) package (v 3.52.4) implemented in the R software environment[63,64] (v 4.2.2) was used to calculate the t-statistics for mean expression values for each gene. The four stages were analysed separately. Surrogate variables were included in the design matrices to further remove any unwanted variations[65]. The read count data fed into the LIMMA linear model fitting were transformed using Voom with quantile normalization[66] followed by group-means parameterization and robust eBayes[64]. The contrast matrix was created with the final goal of identifying the differentially expressed genes in limb that are due to the definite effect of the perturbed growth within each stage. p-values were corrected for multiple testing according to Benjamini and Hochberg (1995) to control the false discovery rate (FDR)[67]. Genes were considered significantly different in expression relative to control if the FDR-adjusted p-value ≤ 0.1. We analysed the expression of canonical molecular signatures from MSigDB[68] using ROAST test from the edgeR package[69] (v 3.38.4), a self-contained, rotation gene set testing which tests whether the genes within a gene set are differentially expressed. The plots were created using ggplot2 package[70] and pheatmap[71] packages from R/Bioconductor and the *Degust* online tool[72].

## Reporting summary

Further information on research design is available in the Nature Portfolio Reporting Summary linked to this article.

# Data availability

The RNA-seq data generated in this study are mapped to the mm39 (GRCm39) reference genome from Ensembl and have been deposited in the GEO repository database under accession code GSE235779. Source data are provided with this paper.

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

## Acknowledgements

We thank Jonathan Gleadle, Vincent Wong and Darling Rojas-Canales for inspiring discussions about mTORC1 and the balance between proliferation and cell size in compensatory responses. We also acknowledge the Monash Bioinformatics Platform (especially Kirill Tsyganov), and Trevor Wilson, at the Medical Genomics Facility, Monash Health Translation Precinct, for their excellent technical help. This study is funded by HFSP CDA00021/2019-C (to A.R-D.) and NHMRC Ideas grant 2002084 (to C.H.H. and A.R-D.). The Australian Regenerative Medicine Institute is supported by grants from the State Government of Victoria and the Australian Government.

## Author contributions

C.H.H.: data acquisition and analysis, supervision, figure preparation and funding sourcing. S.A.: data analysis, manuscript editing, supervision, figure preparation. H.C.: data acquisition and analysis, figure preparation. B.Z: data acquisition, figure preparation. X.Q.: sample processing. D.P.: tool generation, data analysis, supervision. A.R-D.: conceptualization, data acquisition and analysis, funding sourcing, supervision, figure preparation, paper drafting.

## Competing interests

The authors declare no competing interests.
