## [Peer Review File · Nature Communications]

Compensatory growth and recovery of cartilage
cytoarchitecture after transient cell death infetal mouse limbsREVIEWER COMMENTS

Reviewer #1 (Remarks to the Author):

This work by H'ng, et al, Compensatory growth and recovery of tissue cytoarchitecture after transient cartilage-specific cell death in foetal mouse limbs uses genetically elegant mouse models to enact cell death in embryonic cartilage and examine the impact on growth. They also report on mTORC1 signaling impacts on this process.

There are many positive aspects of this report. The authors carefully establish a model wherein substantial, transient cell death is enacted in growing cartilage of long bones (Figure 2). After cell death, substantial repair of the cytoarchitecture is demonstrated (Figure 3), followed by partial catch-up growth (Figure 4).

The resulting shift in EdU incorporation is rigorous (Figure 5), however, it is not entirely clear how to interpret these results with respect to the model introduced in the introduction. Related to this, the introduction is a bit confusing as autonomous growth as a model was not supported in a test from its first strong prediction, leading to a question of why the second one would be tested - it seems the model is already not supported. This study and its results also do not really support it. In the end, this aspect of the story does not really have an ending other than it is complicated.

The mTORC1 pathway and whether the role of this pathway is central to compensatory responses - and what aspect of this pathway - may be critical to the overall response is difficult to conclude in the two figures added to this report.

While not central to the report on the cartilage growth plate, but possibly important to the overall concept of compensatory bone growth, there is a growing number of reports that show chondrocyte behavior and activity is not simply to build a matrix, die and be replaced by bone. It is clear important signaling to osteoblasts is critical. Further, work by the Cheah, Mariani and other groups are showing strong evidence that there may be plasticity/transdifferentiation in the chondrocyte/osteoblast pathway - not directly relevant for the growth zone chondrocytes, but perhaps impacting overall bone growth. At the very least, the authors should acknowledge this dogma shift that is gaining traction.

As an overall assessment, the 'hybrid' model proposed may be true, but doesn't really permit a conceptual leap. It does dispel the autonomous cell hypothesis, which also was not supported by previous tests.

Reviewer #2 (Remarks to the Author):

Appendicular skeletal development is associated with tremendous compensatory growth potential such

that the impact of unilateral deficiencies can be efficiently minimized. This involves what is termed catch-up growth, where through poorly defined mechanisms the affected structure can modulate proliferative and/or differentiation programs to catch-up to the “normal” limb element. Numerous models have been proposed to describe this phenomenon and H’Ng and colleagues have used a sophisticated and elegant genetic model to interrogate the underlying cellular and molecular processes supporting catch-up growth. The model involves selective DTA-mediated ablation of chondrocytes in the left developing skeletal primordia, with the contralateral right limb serving as a control. A combination of approaches have been used to show that catch-up growth appears to involve two stages, an initial lag or stalling of chondrocyte progression, followed by accelerated recovery of chondrocyte differentiation. The studies are well done and robust in nature, however, I have one major concern regarding the model. It is somewhat unexpected why one observes p-S6 induction in both the L and R samples (albeit with different kinetics) in comparison to controls. Similarly, the RNA-seq analyses reveal overlapping patterns of transcript abundance in the L and R samples vs. controls. The authors suggest that a systemic factor may be released following injury of the L limb that impacts the R limb. However, is it plausible that the genetic background and potentially sub-lethal DTA expression may modify the phenotype of cells in both L and R samples.

Comments:

- 1) The introduction is fairly long and I would recommend moving material in lines 87-118 to the discussion.
- 2) As shown in Ext. Data 7, the most notable changes in gene expression appear to be related to the different genotypes, with more subtle difference evident in the left vs right samples (same genotype). It would have been helpful to have RNA-seq data for the rapamycin samples as another measure of potential rescue with this treatment.
- 3) Line 631, “We hypothesize that the cartilage injury is triggering not only a local response but also a systemic one, which includes mTORC1 activation at other sites”. Is this easily testable, perhaps through evaluation of the ribs?
- 4) Line 623, in the experimental group, mTORC1 activity as monitored by p-S6 exhibits different kinetics of induction in the L vs R samples. However, both experimental samples show an appreciable increase in p-S6 in comparison to controls. Similarly, RNA-seq analyses reveal reasonably good concordance between the two experimental groups (L and R) vs. control. As noted in (3 above) and suggested by the authors, this could be the result of a systemic factor that is activated in response to L injury that impacts R. An alternative explanation worth considering (as noted in the narrative above) relates to the genetic background and likely variable DTA expression which may be sufficient to modify cellular phenotypes without lethality.

Minor comments:

- 1) In Figure panel 3b, it would be nice to highlight the deficient areas within the growth plate in the experimental group for the non-expert. Is there a reason, quantification wasn’t carried out for the P0 samples?
- 2) Ext. Data 7c, legend for genotypes doesn't correspond to colors used in the heatmaps.

Reviewer #3 (Remarks to the Author):

This manuscript examines the incredibly interesting phenomenon of “catch-up” growth in developing bone. Work from other researchers in the field have put forth a hypothesis of how growth arrest in developing bone may then be rescued by “catch-up growth” through the autonomous preservation of self-renewal capacity in the chondroprogenitors. The authors test this model using a previously developed a very clever system to perturb chondrocytes unilaterally in the developing growth plate of mouse limb bones. This allows them to compare the chondrocyte dynamics perturbed versus the normal bone. Here they show that sparse killing of chondrocytes in one bone leads to a compensatory self-renewal response in remaining chondroprogenitors, which disproves the previous cell autonomous model. They further show the upregulation of TORC signaling in the remaining cells and the requirement for TORC signaling in catch up growth. This work establishes that damage or removal of chondrocytes in the growing bone can be compensated for non-autonomously by remaining chondroprogenitors which largely but not completely rescue the growth defect.

The work is very carefully performed and quantified and makes an important conceptual point in the field and is therefore appropriate for publication.

Minor comments:

Line 64: When describing the cell autonomous model, compared to their previous work cell cycle arresting cells and seeing a non-autonomous response, the text is a bit confusing because the non-autonomous model is also based on cell cycle arrest of progenitors (that themselves can come out of arrest to rescue the phenotype whereas in the authors own work the progenitors are no longer competent to divide so neighboring cells must compensate). This distinction needs to be made more clear in order to make it evident to the reader why the author’s own work argues for a non-autonomous phenomenon.

Figure 3. Although analyzed with appropriate statistics, would be better if had larger N for all samples.

Reviewer #1 (Remarks to the Author):

This work by H'ng, et al, Compensatory growth and recovery of tissue cytoarchitecture after transient cartilage-specific cell death in foetal mouse limbs uses genetically elegant mouse models to enact cell death in embryonic cartilage and examine the impact on growth. They also report on mTORC1 signaling impacts on this process.

There are many positive aspects of this report. The authors carefully establish a model wherein substantial, transient cell death is enacted in growing cartilage of long bones (Figure 2). After cell death, substantial repair of the cytoarchitecture is demonstrated (Figure 3), followed by partial catch-up growth (Figure 4).

The resulting shift in EdU incorporation is rigorous (Figure 5), however, it is not entirely clear how to interpret these results with respect to the model introduced in the introduction.

We thank the reviewer for these positive comments. Re the EdU results, we agree that it is a complex process to explain. We have added new panels in Fig. 1 in the Introduction, establishing the concept of self-renewal/differentiation balance, so that the EdU pulse-chase is easier to interpret later. We have also slightly modified the description of the experiment to make it clearer (see new lines 867-889 in tracked_changes file). In summary, the pulse-chase results show 2 outcomes: i) During the injury, the replenishment of resting chondrocytes is favoured at the expense of their progression towards proliferative ones, causing an overall delay in bone elongation; ii) after the injury, this balance switches back to favour production of proliferative chondrocytes. Our interpretation is that this initial delay allows the injured cartilage to replenish the progenitor pool, so that the growth potential is not lost as much as it would be if growth just continued normally. This means that instead of shifting the growth curve vertically, as it would be expected in the absence of catch-up growth (see b' in New Figure 1 below), or horizontally, as it would be expected with perfect catch-up growth (see b'' in New Figure 1 below), the growth curve is shifted diagonally, as shown in Figure 8.

Related to this, the introduction is a bit confusing as autonomous growth as a model was not supported in a test from its first strong prediction, leading to a question of why the second one would be tested - it seems the model is already not supported. This study and its results also do not really support it. In the end, this aspect of the story does not really have an ending other than it is complicated.

Thank you for the feedback. We have rewritten the Introduction to improve clarity, moving some parts to the Discussion.

The reason why the cell-autonomous hypothesis still needed to be tested despite our previous work (Rosello-Diez et al. 2018) is that in that work the cell-cycle arrest was continuous, not transient, and therefore a potential role of formerly p21+ chondrocytes could not be followed (see new lines 104-113 in tracked_changes file). By eliminating the targeted cells, our present work allows us to discard cell-autonomous effects and focus on cell-nonautonomous mechanisms. The results are quite different between both injury models, indicating that testing both predictions was indeed worth doing.

We respectfully disagree with the last comment. While we had to decide where to draw the lines in our current study, it is important to highlight that our study provides conclusive findings. The identification of an mTORC1 role in the unexpected response of untargeted cells adds a valuable piece to the puzzle of catch-up growth. Moreover, we have presented ample data supporting a novel hybrid model of catch-up growth, which explains the available data better than previous models. We believe that these contributions mark a significant stride forward in the field, paving the way for further investigations and the exploration of new avenues in research.

The mTORC1 pathway and whether the role of this pathway is central to compensatory responses - and what aspect of this pathway - may be critical to the overall response is difficult to conclude in the two figures added to this report.

Thank you for raising this point. We have further probed the role of mTORC1 by analysing the effect of mTORC1 inhibition on cell size and on the EdU pulse-chase experiments (Fig. 7, Ext. Data Fig. 8 and 9). This has revealed that, at early stages of catch-up growth, mTORC1 is mostly involved in controlling the decision between renewing resting-zone chondrocytes vs. recruiting chondrocytes towards the proliferative zone (i.e., inhibition of mTORC1 led to increased flux of chondrocytes towards the PZ, and reduced retention in the RZ). At late stages of catch-up growth, mTORC1 is involved in maintaining the proliferative potential of PZ chondrocytes, as mTORC1 inhibition led to reduced length of chondrocyte columns, both in normal and catch-up growth.

While not central to the report on the cartilage growth plate, but possibly important to the overall concept of compensatory bone growth, there is a growing number of reports that show chondrocyte behavior and activity is not simply to build a matrix, die and be replaced by bone. It is clear important signaling to osteoblasts is critical. Further, work by the Cheah, Mariani and other groups are showing strong evidence that there may be plasticity/transdifferentiation in the chondrocyte/osteoblast pathway - not directly relevant for the growth zone

chondrocytes, but perhaps impacting overall bone growth. At the very least, the authors should acknowledge this dogma shift that is gaining traction.

The reviewer is right, this has been included in the Introduction (see new lines 59-60 in tracked_changes file).

As an overall assessment, the 'hybrid' model proposed may be true, but doesn't really permit a conceptual leap. It does dispel the autonomous cell hypothesis, which also was not supported by previous tests.

As mentioned above, none of the previous models could account for the results that we have obtained. The hybrid model that we propose is a more likely middle-ground between the total lack of catch-up growth (predicted by the cell-autonomous model), and a hypothetical total recovery (expected if self-renewing progenitors exist and can respond to the loss of transit-amplifying and terminally differentiated chondrocytes). It is therefore a leap in our understanding of developmental robustness and cartilage growth.

Reviewer #2 (Remarks to the Author):

Appendicular skeletal development is associated with tremendous compensatory growth potential such that the impact of unilateral deficiencies can be efficiently minimized. This involves what is termed catch-up growth, where through poorly defined mechanisms the affected structure can modulate proliferative and/or differentiation programs to catch-up to the "normal" limb element. Numerous models have been proposed to describe this phenomenon and H'Ng and colleagues have used a sophisticated and elegant genetic model to interrogate the underlying cellular and molecular processes supporting catch-up growth. The model involves selective DTA-mediated ablation of chondrocytes in the left developing skeletal primordia, with the contralateral right limb serving as a control. A combination of approaches have been used to show that catch-up growth appears to involve two stages, an initial lag or stalling of chondrocyte progression, followed by accelerated recovery of chondrocyte differentiation. The studies are well done and robust in nature, however, I have one major concern regarding the model. It is somewhat unexpected why one observes p-S6 induction in both the L and R samples (albeit with different kinetics) in comparison to controls. Similarly, the RNA-seq analyses reveal overlapping patterns of transcript abundance in the L and R samples vs. controls. The authors suggest that a systemic factor may be released following injury of the L limb that impacts the R limb. However, is it plausible that the genetic background and potentially sub-lethal DTA expression may modify the phenotype of cells in both L and R samples.

Thank you for the accurate and positive summary. Re the contralateral effects that we observed, several compelling results indicate that activation of response pathways in the Exp Right limb are not due to background effects or sublethal DTA expression:

- animals with experimental genetic background (i.e. rtTA+) that are NOT treated with Dox show no trace of ectopic mTORC1 activation in either the Exp L or R cartilage. See Ext Data Fig. 10.

- we could not detect almost any expression of DTA mRNA in Exp R cartilage after Dox treatment. See Extended Data Fig. 1b, c and Extended Data Fig. 10a. Moreover, the extent of mTORC1 activation in the Exp R cartilage is much greater than that of ectopic DTA expression or cell death, strongly suggesting that it is not a direct consequence of DTA expression/cell death.

Comments:

1) The introduction is fairly long and I would recommend moving material in lines 87-118 to the discussion. This has been done, thank you for the suggestion.

2) As shown in Ext. Data 7, the most notable changes in gene expression appear to be related to the different genotypes, with more subtle difference evident in the left vs right samples (same genotype). It would have been helpful to have RNA-seq data for the rapamycin samples as another measure of potential rescue with this treatment.

While we appreciate the suggestion, we think that the uniqueness of our system (i.e. the presence of an internal control within the experimental embryos) allows us to undertake experiments as powerful or more than RNA-seq, at a fraction of the cost. Along these lines, we have addressed the rescue effect by extending the analysis of Rapa-treated samples into all the responses that we explored in the untreated samples (e.g., exacerbated hypertrophy, progenitor replenishment, cartilage architecture), as well as extended it to analyse cell death. See new lines 946-1018 in tracked_changes file. In summary, we found that Rapa treatment did not greatly impact cell death or cell hypertrophy (Ext. Data Fig. 8b and 9c), but did interfere with the replenishment of progenitors, cartilage delay and eventual recovery of cartilage architecture. The fact that the Exp R cartilage did not always respond as the Control cartilage underscores the power of our approach.

3) Line 631, "We hypothesize that the cartilage injury is triggering not only a local response but also a systemic one, which includes mTORC1 activation at other sites". Is this easily testable, perhaps through evaluation of the ribs?

This is a great suggestion, thank you. Although this "distant effect" is not the main focus of this manuscript, but rather the subject of a very preliminary new study, we have undertaken the suggested experiment. Analysis of the rib cartilage, (see Ext. Data Fig. 10b) indeed showed ectopic p-S6 in both left and right experimental ribs, as compared to controls. Future studies will probe this exciting systemic effect further. The main text was modified to reflect this (see new lines 1181-1183 in tracked_changes file).

4) Line 623, in the experimental group, mTORC1 activity as monitored by p-S6 exhibits different kinetics of induction in the L vs R samples. However, both experimental samples show an appreciable increase in p-S6 in comparison to controls. Similarly, RNA-seq analyses reveal reasonably good concordance between the two experimental groups (L and R) vs. control. As noted in (3 above) and suggested by the authors, this could be the result of a systemic factor that is activated in response to L injury that impacts R. An alternative explanation worth considering (as noted in the narrative above) relates to the genetic background and likely variable DTA expression which may be sufficient to modify cellular phenotypes without lethality.

As discussed above, this is very unlikely, as there is almost no detectable DTA expression in the Exp R limb (see Ext. Data Fig. 1c), and the Exp genetic background has no effect on its own (i.e., in the absence of Dox, Ext. Data Fig. 10a).

Minor comments:

1) In Figure panel 3b, it would be nice to highlight the deficient areas within the growth plate in the experimental group for the non-expert. Is there a reason, quantification wasn't carried out for the P0 samples?

We have delineated the damaged zones, as suggested. We quantified the E18.5 instead of the P0 so that it could be directly comparable with the Rapa-treated. Given the uncertainty of the exact delivery time, it is more reproducible to collect samples at E18.5 than P0.

2) Ext. Data 7c, legend for genotypes doesn't correspond to colors used in the heatmaps.

To avoid confusion, we have ensured that the legend colours correspond to the heatmap annotations in the figure 7c (i.e., salmon for controls, green for experimentals, teal for left, lilac for right). Furthermore, we have made the colours match across 7b and 7c.

Reviewer #3 (Remarks to the Author):

This manuscript examines the incredibly interesting phenomenon of “catch-up” growth in developing bone. Work from other researchers in the field have put forth a hypothesis of how growth arrest in developing bone may then be rescued by “catch-up growth” through the autonomous preservation of self-renewal capacity in the chondroprogenitors. The authors test this model using a previously developed a very clever system to perturb chondrocytes unilaterally in the developing growth plate of mouse limb bones. This allows them to compare the chondrocyte dynamics perturbed versus the normal bone. Here they show that sparse killing of chondrocytes in one bone leads to a compensatory self-renewal response in remaining chondroprogenitors, which disproves the previous cell autonomous model. They further show the upregulation of TORC signaling in the remaining cells and the requirement for TORC signaling in catch up growth. This work establishes that damage or removal of chondrocytes in the growing bone can be compensated for non-autonomously by remaining chondroprogenitors which largely but not completely rescue the growth defect.

The work is very carefully performed and quantified and makes an important conceptual point in the field and is therefore appropriate for publication.

Thank you. We appreciate the positive feedback.

Minor comments:

Line 64: When describing the cell autonomous model, compared to their previous work cell cycle arresting cells and seeing a non-autonomous response, the text is a bit confusing because the non-autonomous model is also based on cell cycle arrest of progenitors (that themselves can come out of arrest to rescue the phenotype whereas in the authors own work the progenitors are no longer competent to divide so neighboring cells must compensate). This distinction needs to be made more clear in order to make it evident to the reader why the author's own work argues for a non-autonomous phenomenon.

Thank you for the feedback, we have made this clear now (see new lines 103-113 in tracked_changes file). In summary, the first prediction was tested by us in Rosello-Diez et al. 2018, by inducing continuous mosaic cell-cycle arrest in chondrocytes, as opposed to the transient but widespread arrest induced in the study by Baron et al. 1994. Unexpectedly, the non-arrested chondrocytes showed enhanced proliferation rate, compensating for the proliferative arrest of their neighbours. The most parsimonious interpretation of this result is that the spared chondrocytes showed a cell-nonautonomous response, likely mediated by cell-cell communication and hence not compatible with a strict interpretation of the autonomous model. However, since we could not trace the fate of the arrested chondrocytes, the remote possibility existed that some of them escaped the cell-cycle arrest and participated in the compensation, as suggested by the autonomous model. To avoid the abovementioned caveat and test the autonomous model of catch-up growth in a more definitive way, in this study we killed chondrocytes instead of arresting them, such that they could not possibly participate in the potential recovery.

Figure 3. Although analyzed with appropriate statistics, would be better if had larger N for all samples.

We have expanded the number of samples to at least n=4 in Fig. 3e, in addition to a new time point in Fig. 6b and the new experiments described in the previous answers (e.g. effect of mTORC1 inhibition on cell size and EdU pulse-chase experiments).

REVIEWERS' COMMENTS

Reviewer #1 (Remarks to the Author):

The authors of this work have addressed or clarified the concerns raised in initial review. The hypotheses they set up has been tested in the context of the experiments they report and a clear case is supported that the cells in the growth zone do not operate cell autonomously, but rather communication between cells and the environment occurs in response to injury. They have also showed that mTORC is involved in cell responses.

Reviewer #2 (Remarks to the Author):

The revised manuscript is substantially improved and the authors have adequately addressed my earlier concerns, especially with respect to the model and potentially confounding effects of weak expression of DTA in the right limb.

Reviewer #3 (Remarks to the Author):

The authors have addressed my points raised. This manuscript is certainly worthy of publication in Nature Communications.